

# Why has catchment evaporation increased in the past 40 years? A data-based study in Austria

Doris Duethmann[1], Günter Blöschl[1]

[1]Institute for Hydraulic and Water Resources Engineering, Vienna University of Technology, Karlsplatz 13/223, A-1040 Vienna, Austria.

*Correspondence to*: Doris Duethmann (duethmann@hydro.tuwien.ac.at)

**Abstract.** Global warming has increased regional evapotranspiration in many parts of the world in the last decades, but the drivers of these increases are widely debated. Part of the difficulty lies in the scarcity of high-quality long term data on evapotranspiration. In this paper, we analyze changes in catchment evapotranspiration estimated from the water balances of 156 catchments in Austria over the period 1977–2014 and attribute them to changes in atmospheric demand and available energy, vegetation, and soil moisture as possible drivers. Trend analyses suggest that evapotranspiration has significantly increased in 60 % of the catchments ($p \leq 0.05$) with an average increase of $29 \pm 14$ mm y$^{-1}$ decade$^{-1}$ ($\pm$ standard deviation) or $4.9 \pm 2.3$ % decade$^{-1}$. A pooled pan evaporation series based on 22 stations has, on average, increased by $29 \pm 5$ mm y$^{-1}$ decade$^{-1}$ or $6.0 \pm 1.0$ % decade$^{-1}$. Reference evaporation over the 156 catchments estimated by the Penman-Monteith equation has increased by $18 \pm 5$ mm y$^{-1}$ decade$^{-1}$ or $2.8 \pm 0.7$ % decade$^{-1}$. Of these, 2.1 % are due to increased global radiation and 0.5 % due to increased air temperature according to the Penman-Monteith equation. A satellite-based vegetation index (NDVI) has increased by $0.02 \pm 0.01$ decade$^{-1}$ or $3.1 \pm 1.1$ % decade$^{-1}$. Estimates of reference evaporation accounting for changes in stomata resistance due to changes in NDVI indicate that the increase in vegetation activity has led to a similar increase in reference evaporation as changes in the climate parameters. A regression between trends in evapotranspiration and precipitation, as a proxy of soil moisture, yields a sensitivity of $0.30 \pm 0.04$ mm y$^{-2}$ increase in evapotranspiration to 1 mm y$^{-2}$ increase in precipitation. A synthesis of the data analyses suggests that $38 \pm 13$ % of the observed increase in catchment evapotranspiration can be directly attributed to increased atmospheric demand and available energy, $30 \pm 12$ % to increased vegetation activity, and $32 \pm 5$ % to increased soil moisture due to increases in precipitation.

## 1    Introduction

Evapotranspiration ($E$) is an important process in the water, energy, and carbon cycles and directly controls agricultural productivity and water availability for human purposes. Global warming has increased regional $E$ in many parts of the world in the last decades (Huntington, 2006). However, due to the difficulty of measuring $E$, especially at large spatial scales, the drivers of changing $E$ are still debated.

Decadal changes in catchment $E$ may be inferred from the catchment water balance, as storage changes are usually small over decadal scales. Surprisingly few studies have investigated trends in water balance based evapotranspiration ($E_{wb}$). Those



studies that exist generally found increases in $E_{wb}$ in the 20th century. Examples include large river basins in the US (Milly and Dunne, 2001; Walter et al., 2004; Kramer et al., 2015), the Tibetan Plateau (Zhang et al., 2007), and catchments in Switzerland (Spreafico et al., 2007). A study of 109 basins around the world for the period 1961–1999 found only few significant trends but a tendency towards positive trends in North and South America and Europe and a tendency towards negative trends in

Africa and Siberia (Ukkola and Prentice, 2013).

Observed trends in catchment $E_{wb}$ can be complemented by point measurements, although it is invariably difficult to link point and catchments scales. Lysimeter data from Rietholzbach, Switzerland over 1976–2007 showed a decreasing trend in $E$ in the first half of the period and an increasing trend in the second half (Teuling et al., 2009). Observations based on eddy covariance are usually too short for trend analyses (Wang and Dickinson, 2012), but they have been used to train models that use satellite

and climate data at longer time scales an larger space scales (Jung et al., 2010; Wang et al., 2010; Zhang et al., 2010; Miralles et al., 2014). While these studies generally agree on positive trends since the 1980s, they differ in the magnitude of the estimated trends (Dong and Dai, 2017). It has been noted, however, that the results of such models need to be treated with care as they are sometimes inconsistent with trends from the water balance, particularly for wet basins (Zhang et al., 2012; Liu et al., 2016).

Potential drivers for changes in $E$ are changes in available energy and atmospheric evaporative demand, which is driven by variations in wind, vapor deficit, and air temperature. Available energy and evaporative demand of the atmosphere can be estimated based on climatic drivers or measured with evaporation pans. In many parts of the world (including North America, China, India), annual pan evaporation ($E_{pan}$) has decreased in the second half of the 20th century with rates of 10–40 mm y$^{-1}$ decade$^{-1}$, despite increases in air temperature (Peterson et al., 1995; Roderick et al., 2009; McVicar et al., 2012). In some

instances, this decrease in $E_{pan}$ has been explained by decreases in net radiation and/or wind speed (Roderick and Farquhar, 2002; Roderick et al., 2007). In other instances, decreasing $E_{pan}$ has been interpreted as a consequence of increasing actual evaporation (Brutsaert and Parlange, 1998; Brutsaert, 2013). In Europe, most studies found increasing trends of $E_{pan}$ (e.g. Ireland (1963–2005) (Stanhill and Möller, 2008), England (1957–2004 and 1986–2010) (Stanhill and Möller, 2008; Clark, 2013), Greece (1983–1999) (Papaioannou et al., 2011), and the Czech Republic (1968–2010) (Trnka et al., 2015)). In this

paper, we use the term atmospheric conditions to summarize the drivers available energy and atmospheric demand.

Another potential driver are changes in land cover and vegetation (Piao et al., 2007), including stomata closure caused by increasing atmospheric $CO_2$ concentrations (Gedney et al., 2006). Finally, changes in terrestrial water availability resulting from changes in precipitation may contribute to changing $E$. For example, Jung et al. (2010) attributed the hiatus of the increasing trend in global terrestrial $E$ during 1998–2008 to the limited moisture supply in the southern hemisphere.

Several global scale studies attributed modelled changes in $E$ to their drivers. The land surface models of Douville et al. (2013) could only explain variations in $E$ over 1950–2005 if natural forcings, enhanced greenhouse gas concentrations and aerosols were considered. Based on an ensemble of land surface models that considered variations in climate, land cover, atmospheric





$CO_2$ concentration and nitrogen deposition, Mao et al. (2015) found that $E$ trends over 1982–2013 were dominantly driven by variations in climate, in particular precipitation. Miralles et al. (2014) showed that variations in $E$ in the tropics were strongly influenced by variations in precipitation driven by El Niño/Southern Oscillations. In contrast, using a modified Penman-Monteith equation with detrended input variables, Zhang et al. (2015) concluded that the increase in global terrestrial $E$ over 1982–2013 was largely driven by vegetation greening while changes in global radiation, wind speed, air vapor pressure, air temperature, and atmospheric $CO_2$ concentration had only minor effects. There does not seem to exist a full consensus regarding the drivers of increasing $E$ that has been observed. The above cited studies looked at changes in modeled $E$ at the global scale, based on globally available meteorological data. Complementary to these global studies, there is a need for data-based studies focusing at smaller regions with high quality data.

The aim of this study is to (a) identify changes in catchment evaporation in the past 40 years, and (b) identify the drivers of these changes. We use high-quality data sets of discharge, precipitation, and other climate variables from 156 catchments in Austria during the period 1977–2014. We analyze regional averages over these catchments in order to increase the robustness of the analysis. The potential drivers of changes in catchment evaporation examined are the atmospheric conditions, quantified by reference evaporation ($E_0$) and $E_{pan}$; vegetation, quantified by a satellite-based vegetation index; and soil moisture, quantified by precipitation as a proxy for the available water.

## 2    Data and methods

### 2.1    Water balance data and water balance estimates

#### 2.1.1    Discharge data and catchment attributes

For the analysis of changes in the water balance based evapotranspiration ($E_{wb}$), we identified all catchments in Austria where daily discharge data in the period 1977–2014 (hydrological years, November to October) were available. The beginning of the analysis period was set to 1977 because most discharge series in Austria start in the mid-1970s. Catchments with substantial anthropogenic influences from dams or water withdrawals (Viglione et al., 2013), catchments containing glaciers, and a few high-mountain catchments where observed discharge exceeded observed precipitation were excluded. This selection resulted in a total of 156 catchments (Fig. 1) ranging in size from 23 to 6214 km² (average 316 km²). Land cover was derived from the Corine 2000 data (European Environment Agency, 2016). The land cover is largely dominated by forest and grassland (Table 1). Median catchment elevations were calculated from the SRTM digital elevation model (Jarvis et al., 2008). They range from 287 to 1920 m (average 910 m). All catchments have a PET/P ratio smaller than one and are thus classified as energy-limited or humid according to Budyko (1974). The catchments were assigned to regions with homogeneous variations in climate (Fig. 1), derived from a multi-variable (temperature, precipitation, sunshine duration, air pressure) Principal Component Analysis (Matulla et al., 2003; Matulla, 2005; Auer et al., 2007).





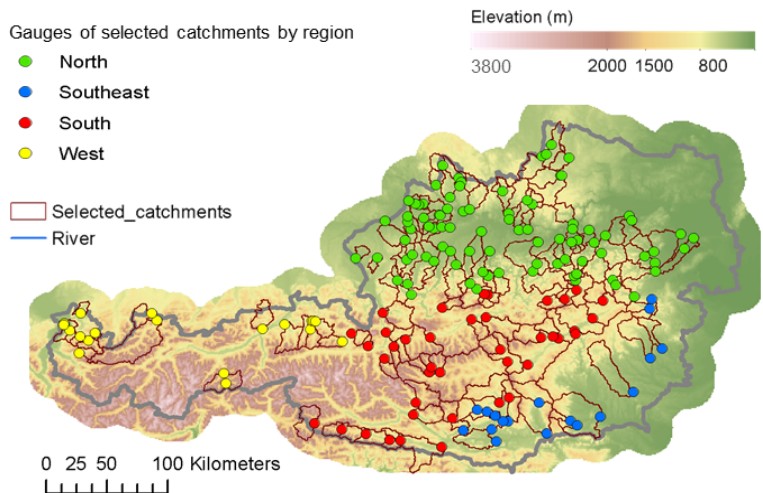

**Fig. 1 Distribution of the study catchments in Austria and their association to one of four regions.**

**Table 1 Characteristics of the 156 study catchments in Austria.**

|  | Median (lower quartile/ upper quartile) |
| --- | --- |
| Area (km²) | 198 (95/368) |
| Median elevation (m) | 825 (571/1218) |
| Coniferous forest (%) | 31 (13/47) |
| Mixed and broadleaf forest (%) | 12 (3/34) |
| Natural grassland (%) | 2 (0/16) |
| Pasture grassland (%) | 12 (6/19) |
| Arable (incl. heterogeneous agricultural areas) (%) | 5 (0/29) |
| PET/P | 0.50 (0.38/0.65) |

### 2.1.2 Catchment average meteorological data

Air temperature and precipitation were obtained from the gridded SPARTACUS data set (Hiebl and Frei, 2016; Hiebl and Frei, 2017). This data set has a temporal and spatial resolutions of 24 h and 1 km, respectively, and was designed to be suitable for trend analyses. The interpolation method of minimum and maximum air temperature accounts for nonlinearities in the thermal profile and uses a constant station network of 150 stations. Precipitation is based on a two-step interpolation scheme, in which 1249 stations (including 119 totalizer precipitation gauges) were used for obtaining a daily background climatology for 1977–2006 and a constant number of 523 stations was used for interpolating ratios between the daily precipitation and the



background climatology. To account for systematic underestimation from gauge undercatch we corrected the gridded precipitation data set for gauge undercatch using the following equation (Richter, 1995)

$$P_{\text{corr}} = P_{\text{orig}} + b \cdot P_{\text{orig}}{}^{e} \tag{1}$$

where $P_{\text{corr}}$ is undercatch corrected precipitation, $P_{\text{orig}}$ uncorrected precipitation, and $b$, $e$ are coefficients that depend on precipitation type and wind exposure. We estimated the precipitation type as snow for mean air temperatures below −1°C, as rain for mean air temperatures above 3°C, and as mixed precipitation between −1°C and 3°C (ATV-DVWK, 2001). The coefficients of Richter (1995) for moderately sheltered locations were applied to all grid points.

Measurements of relative humidity at 7:00 and 14:00 and global radiation were provided by the Austrian Central Institute for Meteorology and Geodynamics (ZAMG). Stations with more than 5 % (15 % for global radiation) missing data during 1977–2014 (hydrological years, November to October) were excluded, which resulted in 125 and 6 stations for relative humidity and global radiation, respectively. Data gaps were filled using linear regression to the station with the highest correlation. The data were interpolated onto a 1 km² grid using local ordinary least squares regression with elevation. The local neighborhood was set to a default radius of 100 km for relative humidity and 200 km for global radiation. This was adjusted to include a minimum of 10 (global radiation 4) and a maximum of 40 stations. The grid values were aggregated to catchment average series. Since wind data were regarded as not representative with respect to evaporation trends (see supplement), uniform monthly wind speeds averaged over all years and over all stations in Austria were used.

### 2.1.3 Estimating evapotranspiration from the water balance ($E_{\text{wb}}$)

The catchment water balance can be written as

$$\frac{dS_{\text{snow}}}{dt} + \frac{dS_{\text{ice}}}{dt} + \frac{dS_{\text{sw}}}{dt} + \frac{dS_{\text{soil}}}{dt} + \frac{dS_{\text{gw}}}{dt} = P - Q - E \tag{2}$$

where $S_{\text{snow}}$ is snow, $S_{\text{ice}}$ ice, $S_{\text{sw}}$ surface water, $S_{\text{soil}}$ soil water, and $S_{\text{gw}}$ ground water storage; $P$ precipitation; and $Q$ discharge. In order to be able to estimate $E$ from the water balance some assumptions on storage changes need to be made. Over periods of several years, we may assume that changes in surface water, soil, and snow storage are small. Studies on groundwater level changes in Austria do not show large-scale groundwater changes over the study period (Blaschke et al., 2011; Neunteufel et al., 2017). Trends in annual mean groundwater levels at 2114 sites in Austria over 1976–2006 showed a heterogeneous picture, with decreasing trends ($p \leq 0.05$) at 18 % of the sites, increasing trends at 12 % of the sites and insignificant trends at 70 % of the sites (Blaschke et al., 2011). We therefore assume that changes in groundwater storage (and changes in any groundwater fluxes) are small. Estimates of absolute values of $E_{\text{wb}}$ (trends in $E_{\text{wb}}$ are not affected) furthermore depend on the assumption that groundwater fluxes across the catchment boundaries can be largely neglected, which is supported by prior rainfall-runoff studies in the catchments that suggest that the water balance can be closed (Parajka et al., 2005). Catchments with glaciers had been excluded from the analysis. For the time scale of decades, catchment evaporation



can therefore simply be estimated as $E_{wb} = P - Q$. Since annual data of $E_{wb}$ should be considered with caution, we applied a Gaussian filter with a standard deviation of two years for the graphical presentation of variations of $E_{wb}$ over the study period.

## 2.2 Pan evaporation data

Daily pan evaporation ($E_{pan}$) data using the GGI-3000 evaporimeter were provided by the Central Hydrographical Bureau
(HZB), Vienna, and from ZAMG, Vienna. Further monthly $E_{pan}$ data were obtained from the Meteorological Yearbook (ZAMG, 1977–1990).The coordinates, elevation, mean, and standard deviation of warm-season $E_{pan}$ are listed in the Appendix (Supplementary Table S1). Missing values in the ZAMG data were replaced by estimates from an empirical Dalton-type formula with locally derived coefficients if wind speed and saturation deficit were available (Neuwirth, 1978). Missing values in the HZB data were replaced by estimates from the climate factor method (Hydrographischen Zentralbüro, 1996) with locally
derived coefficients if wind speed, air temperature, and relative humidity were available. Remaining negative values and values larger than 15 mm d$^{-1}$ were considered erroneous and flagged as missing data. Gaps with a maximum gap size of 4 days were linearly interpolated. Totals over the summer half year (May to October) were calculated if more than 90 % of the daily values or all monthly values were available. Due to the uneven record lengths we analyzed trends for three periods, 1979–2005 (13 stations), 1983–2014 (8 stations), and 1993–2014 (16 stations), which includes only series with a length of at least 20 years
and five years or less missing from the record. Since trend analyses at individual stations did not show regional differences, normalized data of all series with a length of at least 20 years during 1977–2014 (24 stations) were pooled to a common series. The data were normalized by subtracting the mean over the overlapping period 1993–2005, excluding 1995 and 1998, which had many missing values at several stations. Summer $E_{pan}$ was upscaled to the full year by the average ratio of annual and summer reference evapotranspiration (1.33) for comparison.

## 2.3 Estimation of reference evapotranspiration and potential evapotranspiration

### 2.3.1 Reference evapotranspiration

In order to examine effects of changes in atmospheric conditions we estimated reference evapotranspiration ($E_0$) by the Penman-Monteith equation for well-watered short grass vegetation (Allen et al., 1998):

$$E_0 = 0.408 \cdot \frac{\Delta \cdot (R_n - G) + \gamma \cdot \frac{185400}{(T + 273) \cdot r_a} \cdot (e_s - e_a)}{\Delta + \gamma \cdot (1 + \frac{r_s}{r_a})} \tag{3}$$

where $R_n$ is the net radiation at the crop surface (MJ m$^{-2}$ d$^{-1}$), $G$ is the soil heat flux density (MJ m$^{-2}$ d$^{-1}$), $T$ is the mean air
temperature at 2 m height (°C), $r_a$ is the aerodynamic resistance (s m$^{-1}$), $r_s$ is the surface resistance (s m$^{-1}$), $e_s$ is the saturation vapor pressure (kPa), $e_a$ is the actual vapor pressure (kPa), $\Delta$ is the slope of the vapor pressure curve (kPa °C$^{-1}$), and $\gamma$ is the psychrometric constant (kPa °C$^{-1}$). According to the reference conditions of a vegetated surface with a height of 0.12 m, $r_s =$



70 s m$^{-1}$ and $r_a = 208/u_2$ where $u_2$ is the wind speed at 2 m height (m s$^{-1}$). The ground heat flux was neglected. The vapor pressure deficit $e_s - e_a$ was calculated as the average of the vapor pressure deficit at the minimum air temperature (using relative humidity at 7:00) and at the maximum air temperature (using relative humidity at 14:00). $R_n$ was calculated from global radiation ($R_s$; MJ m$^{-2}$ d$^{-1}$), albedo ($\alpha$; set to 0.23) and net longwave radiation ($R_{nl}$; MJ m$^{-2}$ d$^{-1}$)

$$R_n = \alpha \cdot R_s + R_{nl} \tag{4}$$

where $R_{nl}$ was estimated according to Allen et al. (1998) based on minimum and maximum air temperature, clear-sky solar radiation, measured global radiation, and the mean daily vapor pressure. $E_0$ was calculated on a daily basis on a 1 km$^2$ grid and aggregated to catchment average annual values (hydrological years, November to October).

The average contributions of the input variables net radiation, air temperature, and vapor pressure deficit to the trend of $E_0$ were evaluated using estimates of $E_0$ with one or several of the input variables held fixed to a particular year. The year 1994

was selected since annual mean values of $E_0$ and its input variables were close to the mean value over the study period.

The contribution $\varphi_{i,E0}$ of variable $i$ to the trend of $E_0$ in catchment $k$ was calculated as (see e.g., Galbraith et al. (2010)):

$$\varphi_{i,E0}(k) = \frac{\tau_i(k) - \tau_c(k)}{\tau_{E0}(k)} \tag{5}$$

where $\tau_c(k)$ is the trend of the control (where all input variables are kept to those of 1994), $\tau_i(k)$ is the trend of $E_0$ calculated with only variable $i$ varying over the study period (and all other inputs as for the control), and $\tau_{E0}(k)$ is the trend of $E_0$ with all input variables varying. The two-way interaction effect $\varphi_{i\times j,E0}$ of the variables $i$ and $j$ in catchment $k$ was calculated as:

$$\varphi_{i\times j,E0}(k) = \frac{\tau_{i\times j}(k) - \tau_c(k)}{\tau_{E0}(k)} - \varphi_{i,E0}(k) - \varphi_{j,E0}(k) \tag{6}$$

where $\tau_{i\times j}(k)$ is the trend of $E_0$ calculated with variable $i$ and $j$ varying over the study period (and all other inputs as for the control). For average effects and their variability, we calculated averages and spatial standard deviations of $\varphi_{i,E0}(k)$ and $\varphi_{i\times j,E0}(k)$ over all catchments.

### 2.3.2    Effect of changes in vegetation activity on potential evapotranspiration

In order to examine changes in vegetation we used the Normalized Difference Vegetation Index (NDVI) based on satellite

data. Observed 15-day maximum value composite NDVI data at a resolution of 8 km from the Advanced Very High Resolution Radiometer (AVHRR) for 1982–2014 were obtained from Tucker et al. (2005). The NDVI data were aggregated to catchment averages and linearly interpolated to daily catchment average series.

In the Penman-Monteith equation, an increase of the vegetation activity reduces $r_s$, which increases the potential evapotranspiration. We calculated the reference evapotranspiration considering a variable $r_s$ ($E_{0v}$) using Eq. (3), applying a



variable $r_s$ derived from the satellite data instead of a constant $r_s$ of 70 s m$^{-1}$. The vegetation effect was estimated by calculating (i) $E_{0v}$ from the original NDVI series and (ii) $E_{0c}$ from a detrended NDVI series.

We applied two approaches for estimating $r_s$ from NDVI to consider the uncertainty of these estimates. In the first approach, $r_s$ was estimated from the leaf area index (LAI) and the fraction of photosynthetically active radiation (FPAR) which was

estimated from (Sellers et al., 1996):

$$FPAR = \frac{(S - S_{min})}{(S_{max} - S_{min})} \cdot (FPAR_{max} - FPAR_{min}) + FPAR_{min} \qquad (7)$$

Where $S$ is a transformed NDVI value $(1 + NDVI)/(1 - NDVI)$, and $S_{min}$ and $S_{max}$ are the 5 % and 98 % quantiles of $S$ for a given land cover class. LAI was estimated from FPAR (Sellers et al., 1996):

$$LAI = LAI_{max} \cdot \frac{\log(1 - FPAR)}{\log(1 - FPAR_{max})} \qquad (8)$$

where $LAI_{max}$ is the maximum LAI of a land cover class. In Eq. (7) and Eq. (8), we applied the following coefficients for grassland: $NDVI_{min} = 0.039$, $NDVI_{max} = 0.674$, $FPAR_{max} = 0.95$, $FPAR_{min} = 0.001$, and $LAI_{max} = 5$ (Sellers et al., 1996).

$r_s$ was estimated as $r_s = r_l \cdot (LAI \cdot 0.5)^{-1}$ assuming a leaf stomata resistance $r_l$ of 100 s m$^{-1}$ for well-watered grass (Allen et al., 1998).

In the second approach, we used the relationship between $r_s$ and NDVI of Zhang et al. (2010):

$$r_s(NDVI) = \left( \frac{1}{b_1 + b_2 \cdot \exp(-b_3 \cdot NDVI)} + b_4 \right)^{-1} \qquad (9)$$

where $b_1 = 175$ s m$^{-1}$, $b_2 = 2000$ s m$^{-1}$, $b_3 = 6$, and $b_4 = -1/(b_1 + b_2)$ (coefficients for grass).

The average contributions of changes in atmospheric conditions and of changes in vegetation to the trend in $E_{0v}$ ( $\overline{\varphi_{atm,E0v}}$ and

$\overline{\varphi_{veg,E0v}}$) averaged over all catchments and the two approaches for estimating $r_s$ from NDVI were estimated as follows:

$$\overline{\varphi_{atm,E0v}} = \frac{1}{2n} \sum_{k=1}^{n} \sum_{l=1}^{2} \frac{\tau_{E0c}(k,l)}{\tau_{E0v}(k,l)}$$
$$\overline{\varphi_{veg,E0v}} = \frac{1}{2n} \sum_{k=1}^{n} \sum_{l=1}^{2} \frac{\tau_{E0v}(k,l) - \tau_{E0c}(k,l)}{\tau_{E0v}(k,l)} \qquad (10)$$

where $k$ is the catchment index, $n$ the total number of catchments, $l$ refers to one of the two approaches for estimating $r_s$ from NDVI, $\tau_{E0c}(k,l)$ is the trend in $E_{0c}$ and $\tau_{E0v}(k,l)$ is the trend in $E_{0v}$ of catchment $k$ when using approach $l$.



### 2.4 Trend analyses, regression analyses, and attribution of the trend in $E_{wb}$

Trends were estimated by the Sen's slope estimator (Sen, 1968). Trend significance was assessed by the nonparametric Mann-Kendall test (Mann, 1945; Kendall, 1975). The trend free pre-whitening technique was applied to remove lag-one serial correlation (Yue et al., 2002). Uncertainties in the trend magnitude were estimated using a bootstrapping approach. For this

purpose, 1000 samples of size $N$ were drawn, with replacement, from the record of length $N$ years. The Sen's slope was calculated from each of the 1000 samples and the standard deviation was determined. Trends and the standard deviations were first calculated for each catchment and then averaged over the catchments to derive average trends and their uncertainties over a number of catchments.

The trends in $E_{wb}$ in the individual catchments were related to the respective trends in $E_0$, NDVI, and mean annual precipitation

by regression analysis in order to unravel the relation between changes in $E_{wb}$ to changes in atmospheric conditions, changes in vegetation, and changes in soil moisture.

The contributions of the different drivers to the increase in $E_{wb}$ were estimated as follows. The average contribution of changes in soil moisture to the trend in $E_{wb}$ ($\overline{\varphi_{prec,Ewb}}$) was estimated based on the relation between the average trend in $E_{wb}$ and the average trend in annual precipitation:

$$\overline{\varphi_{prec,Ewb}} = \frac{s_{prec,Ewb} \cdot \overline{\tau_{prec}}}{\overline{\tau_{Ewb}}} \tag{11}$$

where $s_{prec,Ewb}$ is the slope of the linear regression of the trend in $E_{wb}$ against the trend in precipitation, and $\overline{\tau_{prec}}$ and $\overline{\tau_{Ewb}}$ are average trends over all catchments in precipitation and $E_{wb}$.

The contributions of changes in atmospheric conditions and vegetation could not be estimated in a similar way since the trends in $E_0$ or in NDVI were not related to trends in $E_{wb}$ (see Sect. 3.2.1 and 0). This is probably due to a relatively low spatial variability in changes in $E_0$ and changes in NDVI. While the spatial variability of changes in precipitation is relatively high,

the spatial variability of changes in available energy, that is an important driver for changes in $E_0$ and in NDVI, is low.

A different approach was therefore used for estimating the average contributions of changes in vegetation and atmospheric conditions to the trend in $E_{wb}$ ($\overline{\varphi_{atm,Ewb}}$ and $\overline{\varphi_{veg,Ewb}}$). Assuming that the remainder of the trend in $E_{wb}$ is caused by changes in atmospheric conditions and vegetation, their contributions were estimated according to the ratio of their effects on $E_{0v}$:

$$\overline{\varphi_{atm,Ewb}} = \left(1 - \overline{\varphi_{prec,Ewb}}\right) \cdot \overline{\varphi_{atm,E0v}}$$

$$\overline{\varphi_{veg,Ewb}} = \left(1 - \overline{\varphi_{prec,Ewb}}\right) \cdot \overline{\varphi_{veg,E0v}} \tag{12}$$

Where $\overline{\varphi_{atm,E0v}}$ and $\overline{\varphi_{veg,E0v}}$ are the contributions of changes in atmospheric conditions and in vegetation to the trend in $E_{0v}$

(see Eq. (10)) averaged over all catchments and both parameterizations for $r_s$.





Uncertainties in the attribution estimate are based on the standard deviation of the regression slope of the trend in precipitation against the trend in $E_{wb}$ and the standard deviations of $\varphi_{atm,E0v}$ and $\varphi_{veg,E0v}$ over all catchments and the two parameterizations for $r_s$.

## 3    Results

### 3.1    Changes in evapotranspiration estimated from the water balance (trend detection)

Catchment $E_{wb}$ trends increased significantly ($p \leq 0.05$) in 93 out of the 156 catchments (60 %) during 1977–2014. One catchment shows a significant decreasing trend. On average over all catchments, the annual $E_{wb}$ increased with a rate of 29 ± 14 mm y$^{-1}$ or 4.9 ± 2.3 % per decade (± standard deviation of the trend; the standard deviation refers to average uncertainties of the trend estimates). The increase was largest during 1980–1995 and flattened out later (Fig. 2a,d). The increase in $E_{wb}$ is more consistent over space and time than the changes in precipitation and discharge (Table 2, Fig. 2b–c, Supplementary Figure S 3), which would be expected and adds credence to the estimates.

Annual precipitation trends increased significantly ($p \leq 0.05$) in 64 out of the 156 catchments (41 %), with an average increase of 32 ± 23 mm y$^{-1}$ or 2.4 ± 1.7 % per decade. Two catchments show significant decreasing trends. Increases were particularly large in the eastern Alpine region of the study domain and generally occurred in summer (Supplementary Figure S 4 and S 5).

**Table 2** Means and trends of catchment evaporation estimated from the water balance ($E_{wb}$), precipitation, and discharge. Numbers given are the spatial averages over regions (and the entire study area) of the mean and the standard deviation.

|  | $E_{wb}$ | | Precipitation | | Discharge | |
|---|---|---|---|---|---|---|
|  | Mean (mm y$^{-1}$) | Trend (mm y$^{-1}$ decade$^{-1}$) | Mean (mm y$^{-1}$) | Trend (mm y$^{-1}$ decade$^{-1}$) | Mean (mm y$^{-1}$) | Trend (mm y$^{-1}$ decade$^{-1}$) |
| North | 625 | 31 ± 13 | 1235 | 38 ± 22 | 610 | 7 ± 21 |
| Southeast | 668 | 27 ± 13 | 1060 | 17 ± 20 | 392 | -8 ± 16 |
| South | 556 | 27 ± 15 | 1411 | 45 ± 25 | 855 | 16 ± 24 |
| West | 546 | 27 ± 19 | 1931 | -13 ± 28 | 1385 | -43 ± 31 |
| All | 604 | 29 ± 14 | 1339 | 32 ± 23 | 735 | 2 ± 23 |



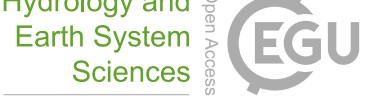

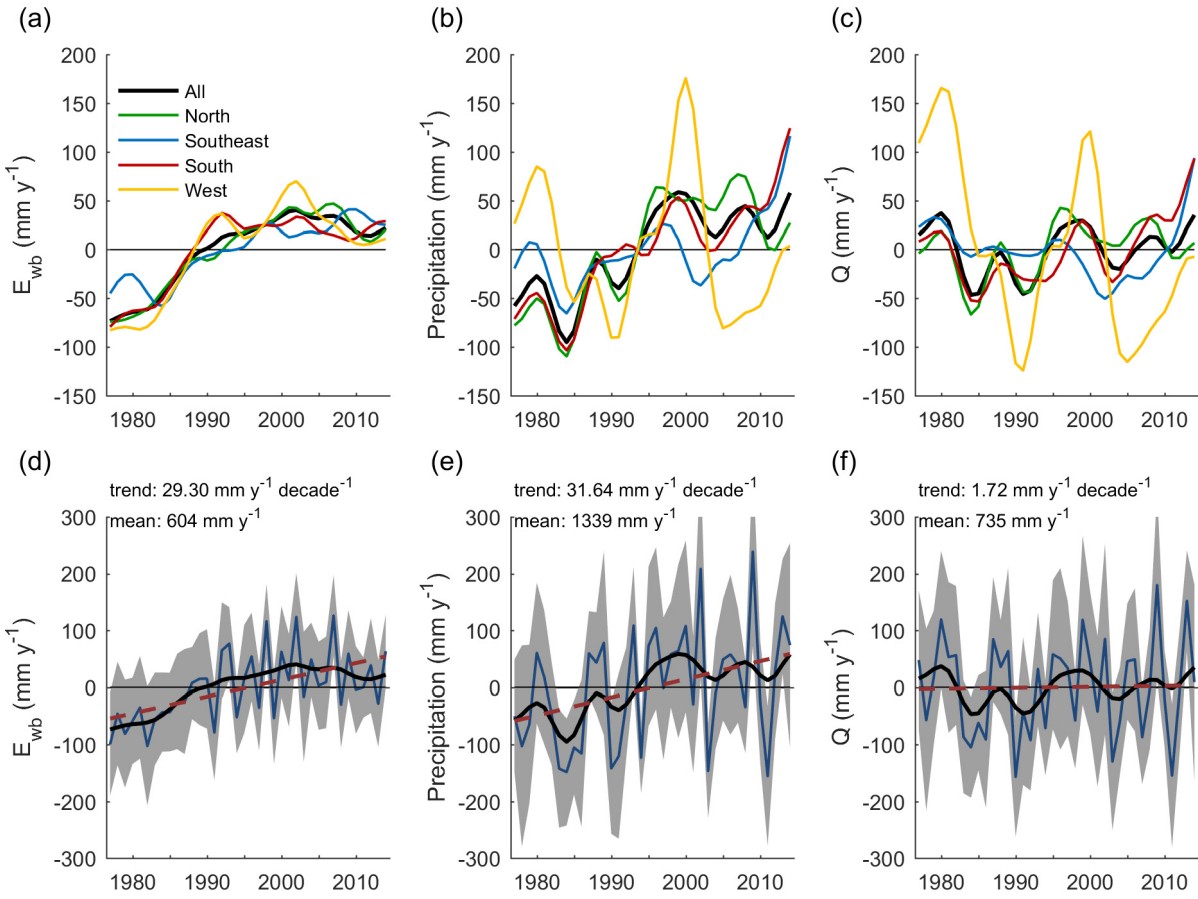

**Fig. 2 Anomalies of (a, d) catchment evaporation estimated from the water balance ($E_{wb}$), (b, e) precipitation and (c, f) discharge over 1977–2014. (a)–(c) mean anomalies by region. Data smoothed using a Gaussian filter with a standard deviation of 2 years. (d)–(f) mean anomalies over all catchments. The thin blue line shows the mean over all catchments, the grey shaded area the variability between catchments (± 1 standard deviation), the bold black line the smoothed mean, and the red dashed line the trend.**

Discharge trends increased significantly ($p \leq 0.05$) in 16 and out of the 156 catchments (10 %) and decreased significantly in 9 catchments (6 %), resulting in an average trend of $2 \pm 23$ mm y$^{-1}$ or $0.2 \pm 3.1$ % per decade. Catchments with increasing discharge are located in the eastern Alpine region where precipitation increased most. Catchments with significant decreasing trends are located in the West of Austria where precipitation did not change much (Supplementary Figure S 3c). Interestingly, the decadal fluctuations of the discharge series within the study period are very similar to those of the precipitation series (Fig. 2b,c).

The analyses in this study are based on undercatch corrected precipitation using coefficients for moderately protected locations. In order to analyze the sensitivity of the correction assumption on the estimated trends in $E_{wb}$, we estimated trends in $E_{wb}$ using uncorrected precipitation and undercatch corrected precipitation with correction parameters for unprotected locations. Without



undercatch correction, estimates of average catchment precipitation would be 9 % lower and the resulting estimates of $E_{wb}$ would be 20 % lower than with undercatch correction for moderately protected locations (Table 3). The percentage of catchments with significant increasing trends in $E_{wb}$ would increase from 60 to 65 %, and the average trend in $E_{wb}$ would increase from 29 to 31 mm y$^{-1}$ or from 4.9 to 6.4 % per decade. Using undercatch correction for wind exposed stations has an effect of similar magnitude but of opposite direction. These results show that, while undercatch correction of precipitation has a strong effect on average $E_{wb}$, it only moderately affects its trends. Please note that in all figures and tables of this paper, with the exception of Table 3, precipitation undercatch has been corrected.

**Table 3 Effect of undercatch correction on estimates of average precipitation ($P$), $E_{wb}$, and their trends (averages over all study catchments and 1977–2014, significant trends for $p \leq 0.05$). *Parameters for moderately protected locations are used for all other analyses in this paper.**

| Undercatch correction | Average $P$ (mm y$^{-1}$) | Percentage of catchments with sign. increases in $P$ (%) | Average increase in $P$ (mm y$^{-1}$ decade$^{-1}$) | Average $E_{wb}$ (mm y$^{-1}$) | Percentage of catchments with sign. increases in $E_{wb}$ (%) | Average increase in $E_{wb}$ (mm y$^{-1}$ decade$^{-1}$) |
|---|---|---|---|---|---|---|
| Parameters for wind exposed locations | 1414 | 35 | 27.1 | 679 | 54 | 25.8 |
| Parameters for moderately protected locations* | 1339 | 41 | 31.6 | 604 | 60 | 29.3 |
| No undercatch correction | 1221 | 45 | 33.7 | 486 | 65 | 31.0 |

## 3.2    Drivers of the increases in evapotranspiration (attribution)

### 3.2.1    Changes in atmospheric conditions – reference evaporation

Changes in the atmospheric conditions were examined by analyzing $E_0$ and $E_{pan}$. Averaged over all catchments, annual $E_0$ increased by $18 \pm 5$ mm y$^{-1}$ or $2.8 \pm 0.7$ % per decade during 1977–2014 (Fig. 3a). Spatial variations in the increase in $E_0$ are small and there is no significant correlation between trends in $E_{wb}$ and trends in $E_0$ ($r^2 = 0.02$, $p = 0.09$) (Fig. 8a). Partial correlations between trends in $E_{wb}$ and $E_0$, when trends in annual precipitation and NDVI were accounted for, are not significant either.

Over our study period, global radiation on average increased by $5.1 \pm 0.9$ W m$^{-2}$ or $3.8 \pm 0.7$ % per decade (Fig. 3b). Maximum air temperature increased by $0.48 \pm 0.11$ °C decade$^{-1}$ (Fig. 3d), while vapor pressure deficit showed variations, with higher values during the late 1990s, but no trend over 1977–2014 (Fig. 3c). The analysis of the $E_0$ estimates with one or several of the input variables held fixed to a particular year showed that the increase in $E_0$ was largely driven by increasing net radiation,



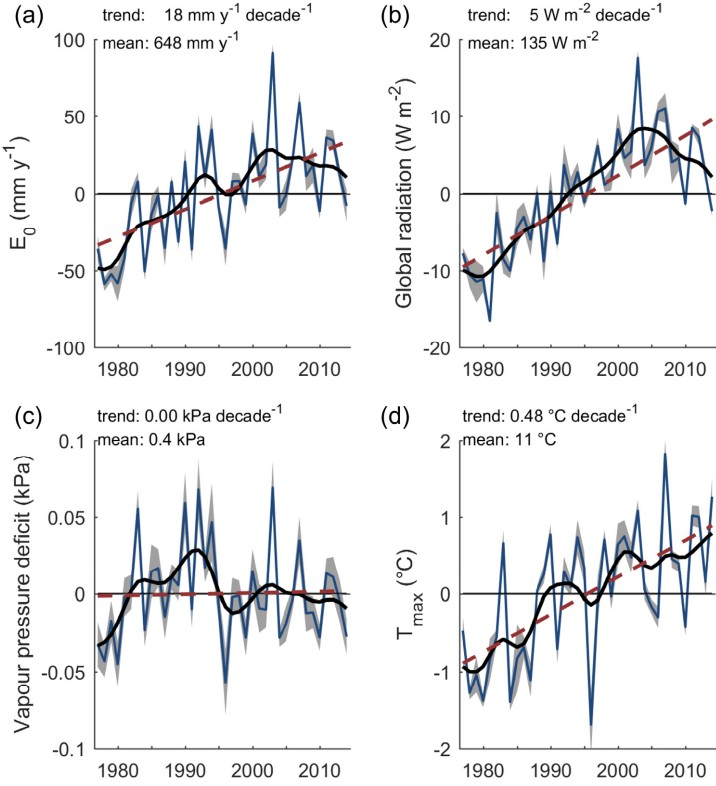

**Fig. 3 Anomalies of (a) $E_0$, (b) global radiation, (c) vapor pressure deficit, (d) maximum air temperature over 1977–2014 and all study catchments. The thin blue line shows the mean over all catchments, the grey shaded area shows the variability between catchments (± 1 standard deviation), the bold black line shows the filtered mean (10-year Gauss filter with a standard deviation of 2 years), and the dashed red line the linear trend.**

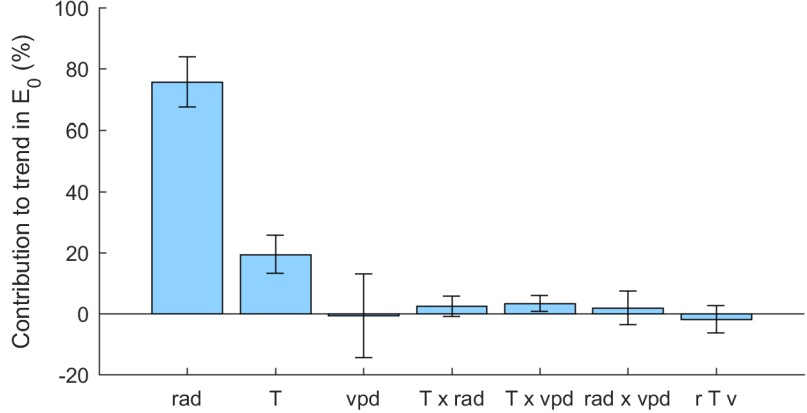

**Fig. 4 Mean contributions of variations in net radiation (rad), air temperature (T) and vapor pressure deficit (vpd) and their two-way and three-way interaction effects to the trend in $E_0$. Bars show means over all catchments, error bars show the standard deviation of the variation between catchments. Percent are relative to trends in $E_0$.**



which contributed $76 \pm 8$ % to the trend in $E_0$, and increases in air temperature, which contributed $19 \pm 6$ % to the trend in $E_0$ (Fig. 4).

### 3.2.2 Changes in atmospheric conditions – pan evaporation

Pan evaporation significantly ($p \leq 0.05$) increased at 5 of 13 stations over 1979–2005, at 4 of 8 stations over 1983–2014, and at 5 of 16 stations over 1993–2014; there are no significant negative trends (Supplementary Figure S 6, Supplementary Table S1) and the regional differences in the trends are small. The average normalized $E_{pan}$ series shows a highly significant ($p \leq 0.01$) increase of $29 \pm 5$ mm y$^{-1}$ or $6.0 \pm 1.0$ % decade$^{-1}$ over 1977–2014 (Fig. 5).

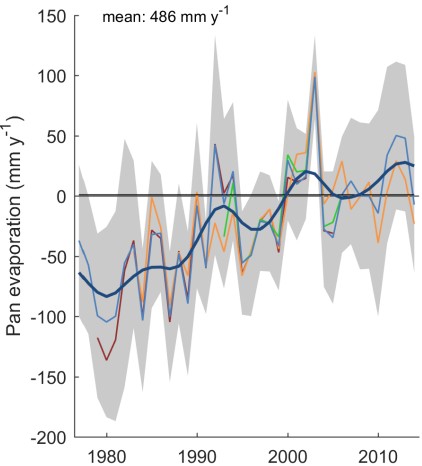

**Fig. 5 Pan evaporation anomalies of 25 stations with a minimum of 20 years available data during 1977–2014. Thin blue line: mean, grey area: ± 1 standard deviation, thick blue line: filtered mean (Gauss filter with a standard deviation of 2 years). The red (orange, green) lines show means for subsets of stations for 1979–2005 (1983–2014, 1993–2014; maximum of 5 years missing during the period).**



### 3.2.3 Changes in vegetation activity

Catchment average NDVI shows a clear seasonal cycle with high values in summer and low values in winter (Fig. 6a, Supplementary Figure S 7). Mean trends in NDVI over all catchments for 15-day composites are positive nearly over the entire year and particularly strong during March/April and November/December (Fig. 6b). Timing of the trends is correlated to

median catchment elevation. During October–March, positive trends are mostly observed in catchments with low median elevation, while during May–June, this reverses and stronger positive trends are observed in high elevation catchments (Supplementary Figure S 8). The average and standard deviation over all catchments of the trend in the average annual NDVI is $0.02 \pm 0.01$ decade$^{-1}$ or $3.1 \pm 1.1$ % decade$^{-1}$.

To estimate the effect of these vegetation changes on $E$ we calculated $E_{0v}$ with $r_s$ estimated based on the original observed

NDVI data and $E_{0c}$ with $r_s$ based on detrended NDVI data. $E_{0v}$, which reflects changes in vegetation activity and atmospheric conditions, showed a stronger increase over the study period than $E_{0c}$, which reflects changes in atmospheric conditions only (Fig. 7a, b). Estimated as average and standard deviation over all catchments and over both approaches for estimating $r_s$, the contribution of changes in atmospheric conditions to the trend in $E_{0v}$ was $56 \pm 15$ % and the contribution of changes in vegetation to the trend in $E_{0v}$ was $44 \pm 15$ % (Fig. 7c).

Changes in annual cumulative NDVI are not correlated to changes in $E_{wb}$ ($r^2=0.01$, $p=0.23$) (Fig. 8b). This was also the case for partial correlations between trends in $E_{wb}$ and NDVI, when trends in annual precipitation or trends in $E_0$ were taken into account.

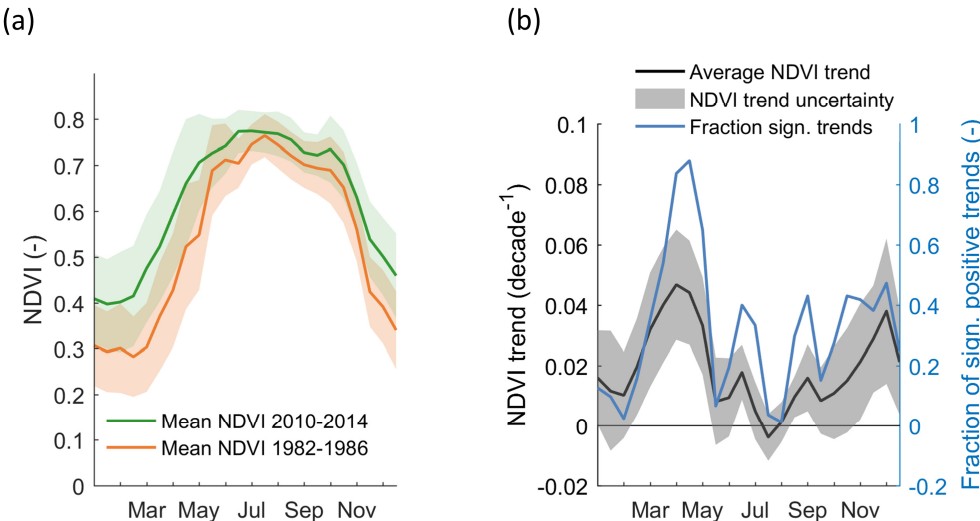

**Fig. 6 Changes in NDVI. (a) Seasonal cycle of NDVI averaged over 1982–1986 and 2010–2014. Solid lines show averages over all**
**catchments and shaded areas show ± 1 standard deviation. (b) Seasonal cycle of trends in catchment average NDVI values over**
**1982–2014. The solid line shows the mean NDVI trend over all catchments, the grey shaded area the spatial variability of the trends**
**between catchments (± 1 standard deviation), and the blue line the fraction of significant positive trends ($p \leq 0.05$).**





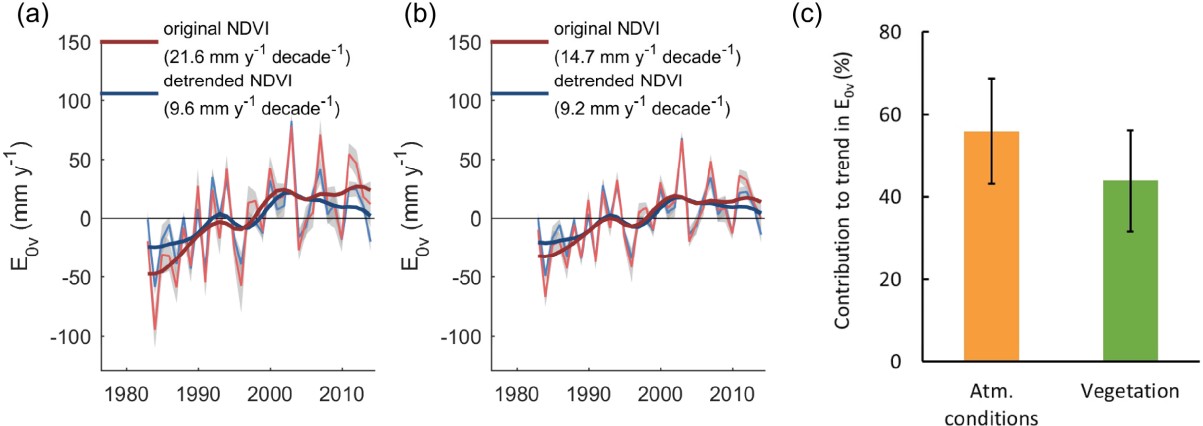

**Fig. 7** Effect of variations in atmospheric conditions and vegetation on $E_{0v}$ (a, b) Anomalies of $E_{0v}$ with $r_s$ estimated with original and detrended NDVI data, (a) $r_s$ estimated based on Sellers et al. (1996), (b) $r_s$ estimated based on Zhang *et al* (2010). The thin line shows the mean over all catchments, the grey shaded area shows the variability between catchments (± 1 standard deviation), and the thick blue line shows the filtered mean (Gauss filter with a standard deviation of 2 years). The number in brackets gives the trend estimate over 1982–2014. (c) Contributions of variations in atmospheric conditions and variations in vegetation to the trend in $E_{0v}$. Bars show means over all catchments and over both approaches for estimating $r_s$, error bars show the variability over all catchments and over both approaches for estimating $r_s$ (± 1 standard deviation). Percent are relative to trends in $E_{0v}$.

### 3.2.4    Changes in soil moisture

We used annual precipitation as a proxy for the water available for $E$. Trends in annual precipitation are described in Sect. 3.1. Higher increases in $E_{wb}$ are observed in catchments with higher increases in annual precipitation ($r^2=0.24$, $p<0.0001$; Fig. 8c). A linear regression of the trend in $E_{wb}$ against the trend in annual precipitation suggests that 1 mm $y^{-2}$ increase in precipitation is associated with $0.30 \pm 0.04$ mm $y^{-2}$ increase in $E_{wb}$. Thus, with an average precipitation trend of 32 mm $y^{-1}$ decade$^{-1}$, on average $9.4 \pm 1.4$ mm $y^{-1}$ decade$^{-1}$ (uncertainty relates to standard deviation of the trend slope) of the $E_{wb}$ trend may be related to the increase in precipitation.

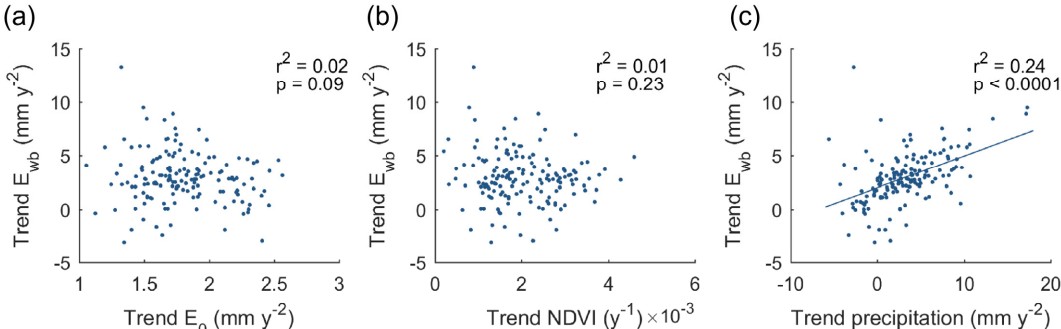

**Fig. 8** Scatter plots of trends in (a) $E_0$, (b) NDVI and (c) annual precipitation against the trend in $E_{wb}$.





### 3.3  Synthesis of attribution

We may now estimate the contributions of the different drivers to the increase in $E_{wb}$. From the regression of the trend in $E_{wb}$ against the trend in annual precipitation (Fig. 8c) we found that, on average, $9.4 \pm 1.4$ mm y$^{-1}$ decade$^{-1}$ of the $E_{wb}$ trend of 29.3 mm y$^{-1}$ decade$^{-1}$ are related to the increase in precipitation (Sect. 3.2.4). The relative contributions of atmospheric conditions and vegetation were assumed to conform to their relative effects on $E_{0v}$ (Sect. 3.2.3, Fig. 7c). Thus the remaining $19.9 \pm 1.4$ mm y$^{-1}$ decade$^{-1}$ are split at a ratio of $0.56 \pm 0.15$ to $0.44 \pm 0.15$ into being due to atmospheric conditions and vegetation, respectively. This results in a contribution of changes in atmospheric conditions of $11.1 \pm 3.8$ mm y$^{-1}$ decade$^{-1}$ and in a contribution of changes in vegetation of $8.8 \pm 3.6$ mm y$^{-1}$ decade$^{-1}$. In summary, the data suggest that changes in atmospheric conditions, vegetation activity, and soil moisture have contributed $38 \pm 13$ %, $30 \pm 12$ %, and $32 \pm 5$ %, respectively, to the average increase in $E_{wb}$ in the study catchments (Fig. 9).

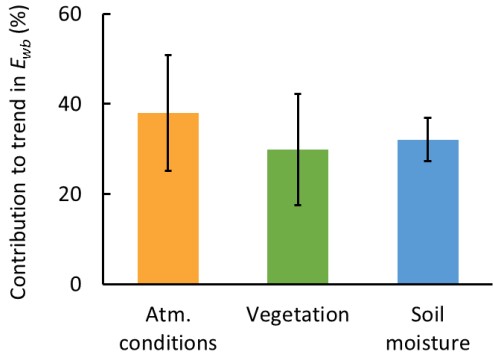

**Fig. 9 Average contributions to the average trend of catchment evaporation estimated from the water balance ($E_{wb}$) from changes in atmospheric conditions, vegetation activity, and soil moisture. Error bars relate to the standard deviation of the estimate. Percent relate to the total average trend of $E_{wb}$ of 29 mm y$^{-1}$ decade$^{-1}$.**

## 4  Discussion

### 4.1  Changes in catchment evapotranspiration in Austria over the past four decades

Based on the analysis of the water balances, we found an average $E_{wb}$ increase of $29 \pm 14$ mm y$^{-1}$ decade$^{-1}$, i.e. a total increase of $108 \pm 52$ mm y$^{-1}$ or $17.9 \pm 8.5$ % over 37 years. This increase is consistent between the different regions, which points towards the importance of drivers with spatially consistent changes across the study region, i.e. global radiation or air temperature rather than changes in precipitation. The increase is strongest in the beginning of the study period and seems to have stopped around 2000. A similar pattern is observed for $E_0$ and $E_{0v}$ (Fig. 3a and Fig. 7a,b) where the decreasing tendency in global radiation (Fig. 3b) appears to be the main cause for the decreasing tendency after 2000.





The $E_{wb}$ estimates could potentially be influenced by changes in groundwater storage. Changes in groundwater storage were assumed to be small over time scales of decades. This assumption is supported by the absence of general trends in discharge, together with the incoherent picture of trends in groundwater levels (Blaschke et al., 2011; Neunteufel et al., 2017), which makes it unlikely that groundwater storage changes have a big influence on the $E_{wb}$ estimates.

Increases in $E_{wb}$ of a similar magnitude have also been observed for Switzerland, where $E_{wb}$ increased by ~20 mm y$^{-1}$ decade$^{-1}$ over 1977–2007 (Spreafico et al., 2007, from Figure 1 on plate 6.6). Lower rates of increase in $E_{wb}$ were observed in other regions and other periods. Estimates based on the water balance were $10 \pm 5$ mm y$^{-1}$ decade$^{-1}$ in several large catchments across the conterminous US during 1950–2000 (Walter et al., 2004), $6 \pm 4$ mm y$^{-1}$ decade$^{-1}$ in catchments in the eastern US during 1901–2009 (Kramer et al., 2015), and 7 mm y$^{-1}$ decade$^{-1}$ in 16 catchments on the Tibetan Plateau during 1966–2000

(Zhang et al., 2007). The larger increases of $E$ in our study may be related to the large increases in air temperature and global radiation in the study region over the period considered.

## 4.2   Drivers of the observed changes in catchment evapotranspiration

All three drivers investigated – changes in atmospheric conditions, changes in vegetation activity and changes in soil moisture – are found to be important for the observed increase in catchment evapotranspiration and the direct effects of these drivers on

$E$ are in a similar order of magnitude. The drivers are closely interlinked. For example, increases in air temperature and precipitation not only contribute directly to changes in $E$, but also indirectly through increases in the vegetation activity. Attributing changes in $E$ to the drivers can therefore be done in different ways. While, in this study, vegetation effects on $E$ include land use changes and indirect effects via climate and atmospheric $CO_2$ on vegetation, other studies have treated the indirect effects separately. Using a global biosphere model, Piao et al. (2007) attributed a global change in $E$ of +0.3 mm y$^{-1}$

decade$^{-1}$ during 1901–1999 to climate (+0.7 mm y$^{-1}$ decade$^{-1}$; including indirect effects of climate on vegetation, atmospheric conditions and precipitation) and land use changes (−0.8 mm y$^{-1}$ decade$^{-1}$; mainly deforestation). Changes in atmospheric $CO_2$ had a further positive effect of +0.4 mm y$^{-1}$ decade$^{-1}$ through increasing LAI and stomata resistance (Piao et al., 2007). Based on global land surface model simulations, Mao et al. (2015) found that climate effects (including indirect effects of climate on vegetation, atmospheric conditions and precipitation) on $E$ were larger than the effects of land use, atmospheric $CO_2$ and

nitrogen deposition during 1982–2013, with precipitation being the most important climate variable. The latter finding is in line with several other studies (Jung et al., 2010; Miralles et al., 2014) although Zhang et al. (2015) found vegetation change (including indirect effects of climate on vegetation) to be more important.

In this study, increases in $E$ driven by changes in atmospheric conditions have been identified by increases in $E_0$ and $E_{pan}$. The main driver for the increase in $E_0$ in our study is an increase in global radiation of on average $5.1 \pm 0.9$ W m$^{-2}$ decade$^{-1}$ with a

further contribution from an increase in air temperature of $0.48 \pm 0.11$ °C decade$^{-1}$. Teuling et al. (2009) and Wang et al. (2010) reported a strong influence of global radiation on changes in $E$ in Europe. Global radiation in Europe generally decreased during the 1960s–1980s ('global dimming') and increased from the 1980s ('global brightening') (Norris and Wild,





2007; Wild, 2009; Sanchez-Lorenzo et al., 2015). The magnitude of the increase in the present study is slightly higher than those reported in other studies ($5.1 \pm 0.9$ W m$^{-2}$ decade$^{-1}$ in this study; and $2.0 \pm 1.2$ W m$^{-2}$ decade$^{-1}$ in Central Europe during 1971–2012 found by Sanchez-Lorenzo et al. (2015) and 3.7 W m$^{-2}$ decade$^{-1}$ in Austria/Switzerland during 1985–2005 found by Wild et al. (2009)). The increases in global radiation over Europe since the mid 1980s have mainly been attributed to the
reduction in aerosols (Norris and Wild, 2007). The strong sensitivity of $E_0$ to global radiation suggests that projected air temperature increases do not necessarily imply increased evaporation in the future.

The observed increase in $E_{\text{pan}}$ in our study is consistent with other European studies. In Greece, non-significant increases in $E_{\text{pan}}$ over 1983–1999 were observed for a pooled series from 14 stations (Papaioannou et al., 2011). Three out of eight sites in Ireland showed a significant increase in $E_{\text{pan}}$ over 1963–2005, and one site showed a significant decrease (Stanhill and Möller,
2008). In England, significant increases in $E_{\text{pan}}$ were observed at two stations over 1957–2004 and 1986–2010, respectively (Stanhill and Möller, 2008; Clark, 2013), and in the Czech Republic April–June $E_{\text{pan}}$ significantly increased at three out of five stations during 1968-2010 (Trnka et al., 2015).

The NDVI data shows a marked increase in vegetation activity in Austria, similar to many other studies in the Northern Hemisphere (Myneni et al., 1997; Slayback et al., 2003; Liu et al., 2015). Increases in air temperature and precipitation and
$CO_2$ fertilization have been identified as drivers (Piao et al., 2006; Los, 2013). We found the strongest increases in NDVI in spring and autumn, indicating a lengthening of the active growing season which has, e.g., been noted by Myneni et al. (1997). Increases in NDVI may further be enhanced by land cover changes. In Austria, forest area increased from 43 % to 47 % over 1977–2010 at the expense of cropland and extensive grassland (Krausmann et al., 2003; Gingrich et al., 2015) as agricultural land in remote areas with low productivity was abandoned due to economic pressure (Tasser and Tappeiner, 2002; Rutherford
et al., 2008).

According to this study the effect of increased vegetation activity on $E_{0v}$ is of a similar magnitude as the effect of changes in the atmospheric conditions. A strong influence of changes in vegetation activity on $E$ was also suggested for the eastern US based on correlations between NDVI and $E_{\text{wb}}$ (Kramer et al., 2015). Using simulations from a coupled biosphere-atmosphere model, Bounoua et al. (2000) found an $E$ increase of 43 mm y$^{-1}$ for an NDVI increase of 0.08, which is slightly higher than
the average NDVI increase observed in this study of 0.06. This study, however, does not account for the effect of increasing atmospheric $CO_2$ concentrations on increasing stomata resistance, which means that the effect of vegetation might be overestimated. Stomata closure due to increased atmospheric $CO_2$ may have reduced global $E$ since 1960 by 1.6 to 2.0 mm y$^{-1}$ decade$^{-1}$ (Gedney et al., 2006; Piao et al., 2007).

Even though the study area can generally be classified as humid, we found a strong influence of changes in precipitation on $E$.
While $E$ is generally energy-limited in the study region (Sect. 2.1.1), $E$ may frequently be limited by available moisture, particularly interception and $E$ from the soil and non-vegetated areas, so that an increase in soil moisture can lead to an increase in $E$ (Parajka et al., 2007; Parajka et al., 2009). At the global scale and in particular in the tropics, several studies found that





changes in precipitation were the most important factor for changes in $E$ (Jung et al., 2010; Miralles et al., 2014; Mao et al., 2015).

## 5    Conclusions and implications

Over the past four decades (1977–2014), catchment evapotranspiration increased on average over 156 study catchments in Austria by $29 \pm 14$ mm y$^{-1}$ decade$^{-1}$. This increase was attributed with similar orders of magnitude to changes in atmospheric demand and available energy (that accounted for $11.1 \pm 3.8$ mm y$^{-1}$ decade$^{-1}$), changes in vegetation ($8.8 \pm 3.6$ mm y$^{-1}$ decade$^{-1}$), and changes in soil moisture ($9.4 \pm 1.4$ mm y$^{-1}$ decade$^{-1}$).

$E_0$ increased on average over all study catchments by $18 \pm 5$ mm y$^{-1}$ decade$^{-1}$. The increase in $E_0$ was largely driven by the increase in global radiation with further contributions from increasing air temperature. Rising atmospheric demand and energy available for $E$ was also revealed by increases in the available $E_{pan}$ data. Satellite-derived NDVI data for 1982–2014 indicate an increase in vegetation activity. This increase may have led to a similar increase of $E_{0v}$ as the climate variables. A positive correlation between increases in $E$ and increases in precipitation furthermore points to increases in soil water availability as a third driver for the increases in $E_{wb}$.

Over the study period, trends in annual discharge were close to zero and increases in $E$ were balanced by increases in precipitation. If the increase in precipitation had been lower and the increase in $E$ had been similar as in this study, notable reductions in discharge would have been likely. A lower increase in precipitation would likely have reduced the increase in $E$ as about a third of the $E$ increase was directly attributed to the increase in precipitation. Furthermore, the increase in $E$ due to the increase in vegetation activity might have been slightly lower without the increase in precipitation.

Estimates of future changes in $E$ of climate impact assessments are often based on predicted air temperature and precipitation changes. This study clearly shows that, despite the large air temperature increase in the recent decades, global radiation was much more important for changes in $E_0$ than air temperature. Climate impact studies on $E$ should therefore explicitly account for possible changes in global radiation. Furthermore, hydrologic models used in such studies should consider the effects of possible changes in vegetation on $E$ that for example result from a longer growing period.

### Acknowledgements

We gratefully acknowledge financial support from the DFG (German Research Foundation) through a Research Scholarship to D.D. (DU 1595/1-1). We would like to thank the Austrian Hydrographic Services and the ZAMG for providing the hydrographic and meteorological data and Juraj Parajka for useful suggestions on processing the data.



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
