# Peer review of "Why has catchment evaporation increased in the past 40 years? A data-based study in Austria"

_Hydrology and Earth System Sciences, 2018_

## Short Comment (SC1) · 12 Apr 2018

In my opinion this one of the few articles that try to look at evapotranspiration variability from various perspectives and therefore is a particularly valuable contribution. Thus said I regret very much to find a point that I would see as a quite fundamental flaw in the analysis.

There is a vast body of evidence (see in particular the review paper of McVicar et al. (2012, cited) that both radiation and aerodynamic terms are determining potential evapotranspiration (Ep). Depending on region wind may account for the major part of Ep variance. Even though the authors argue with a lack of spatial homogeneity of wind

speeds in their study area their averaged wind speeds (Supplement Figure S2b) show quite well the general decrease of wind speeds that has been observed world-wide. The large variability is to be expected in largely mountainous Austria and disregarding this variability does introduce a major error into the analysis. In this respect it is most unfortunate to see that both calculation AND attribution analysis of Penman-Monteith Ep are based on spatially and temporally averaged wind data. Even using averaged wind data as additional variable in the attribution analysis would already show that both radiative and aerodynamic forcing largely explain most of the variance. In the present form – without a realistic inclusion of wind data – the results are misleading. I would propose to recalculate Ep with wind speed data that contains as much spatial and temporal variance as possible. In addition attribution itself is variable both in spatial and temporal terms (Fan and Thomas, https://doi.org/10.1016/j.jhydrol.2018.02.080) so an extended analysis taking into account attribution variability would offer the reader a considerably improved analysis of Ep dynamics.

There are two smaller points I would also like to mention: the authors rightly point out in their paper that temperature is not an important driver of Ep but at the same time begin their introduction with the much-too–often-heard statement that global warming (hence temperature) has increased regional evapotranspiration. Even the IPCC still voices this scientifically incorrect statement. I would propose to rephrase this sentence to clarify that in the context of global CLIMATIC change Ep also has seen changes.

Another point to clarify is the sometimes misleading way 'evapotranspiration' is used in this paper. Evapotranspiration is an umbrella term that has many definitions and can be estimated in different ways. In the Introduction most of the papers cited deal with actual evapotranspiration as do the authors when they use the abbreviation 'E' in their data analysis and results. On p 2/l 15 however they appear to mean potential evapotranspiration (at least most of the cited papers deal with Penman-Monteith potential evapotranspiration). On p 3/l 27 it is PET (again perhaps potential evapotranspiration) while reference evapotranspiration (E0, p 6/l 22) is actually crop reference evapotran-

spiration as the method of Allen et al. 1998 is cited. 'potential evapotranspiration' is used twice in section headlines 2.3 and 2.3.2 but is not defined elsewhere; on p 7/l 23ff 'potential evapotranspiration' and 'reference evapotranspiration' are used almost synonymously. Perhaps the authors might consider adding a short section pointing out the differences between different measures and methods of evapotranspiration cited or used in their paper and then use the appropriate terms consistently throughout their paper.

---

## Referee Comment (RC1) · Dr. Teuling (Referee) · 29 Apr 2018

The manuscript by Duetchmann and Blöschl discusses recent trends in estimates of evapotranspiration in Austria. The analysis is mostly observation-based, and makes use of a range of long-term observations ranging from station- to satellite-data. The work is important because it is one of the first thorough and detailed investigations on trends in ET in a region that has been identified by other studies as a region where trends in global radiation have been significant, and where they would likely have impacted ET. This study partly confirms these previous findings, but also shows that other factors such as vegetation might also affect ET trends. In general the findings are robust and well-presented. My main concerns, partly in line with the comments by the first referee, lies with the interpretation and quantification of the contributions to the ET trend (Figures 7c and 9).

Concerning the effect of wind: the potential impact should be discussed in more detail. Wind speed is known to have seen significant trends in many regions (Vautard et al., Nature Geosci. 3, 756-761), and this could have impacted ET trends also in this study. According to the offline PM-equation used by the authors, the impact of wind is direct. It should be noted, however, that when coupled to an atmospheric model, the sensitivity of PM ET for wind becomes much smaller (see e.g. Van Heerwaarden et al., Geophys. Res. Lett. 37, L21401). So I consider it unlikely that wind is a strong driver of ET trends locally, but I agree with the other referee that this warrants an in-depth discussion.

My main concern related to the interpretation of Figure 8c. This figure shows the relation between inferred trends in P and ET. It suggests a very strong control of P on ET trends, which seems somewhat suspicious given the general humid climate conditions in Austria. In my view, two possible explanations exist. It might be that in general, soil moisture constraints on ET have weakened because of increased P. In this case, one would expect inferred actual ET to be significantly lower than potential ET. This relation between actual and potential ET, however, is not explored in the manuscript. I believe such an analysis should be included in a revised version, as it provides important insight into the possible background of the drivers of ET trends. It should be noted that trends in soil moisture and vegetation might possibly be related. This needs to be discussed, along with the implications for the results shown in Figure 9 which assumes soil moisture/P and vegetation effects to be independent. A second explanation for the relation in Fig. 8c could be that trends in ET are induced by overestimation of trends in P, for instance due to a too strong correction for undercatch. This possibility needs to be explored and discussed. So in summary, if the correlation is physical/causal, the authors should provide additional evidence for the underlying process, for instance by showing increasing ET/PET ratios. In addition, the dependency of trends in vegetation and soil moisture needs to be explored. Fig 9 is interesting, but these results are currently not sufficiently robust to be published.

---

## Referee Comment (RC2) · Anonymous Referee #2 · 30 Apr 2018

To fully appreciate this paper, one needs to first consider adopting the traditional approach to studying climate change impacts on long-term water balances. Say one builds a model of long-term water balance – with fixed parameterizations of runoff and evapotranspiration as functions of soil moisture. Then calibrate the model with past runoff and then impose some kind of future climate and make a prediction of what the water balance will be like in the future under the changed climate. The main message coming out of this paper is the fallacy of this kind of approach. I believe that this paper is an important milestone in the study of climate change impacts on catchment water balances.

[Figure]

The paper focuses on long-terms changes in evapotranspiration across Austria based on water balance analyses using observed data on P and Q: and estimation of E as the residual P-Q (assuming negligible ïĄŠS, which they justify based on measured groundwater levels across Austria). The main outcome of the water balance study is that there has been a significant increasing trend in ET (water balance based estimate) over the past 40 years, even as there is no significant trend in Q over the same period. This is intriguing, although the authors say this has been observed in other places too. Why would ET increase and yet Q would remain the same?

The authors carry out a detailed attribution study to discover the possible causes of this phenomenon. The analyses lead them to conclude that the increase in ET is attributable to 3 apparent causes, in almost equal measure: atmospheric conditions (radiation, temperature), vegetation activity (i.e., NDVI) and precipitation. No problems so far for me. Under atmospheric conditions VPD was looked at and did not have much of an impact: a reviewer questioned why wind speed was not looked at – hopefully the authors would clarify that.

The authors explain the dependence of the ET change (partly) on change of precipitation is as a result of increased soil moisture arising from the precipitation change. Here is where I have some concern - I have trouble grasping the attribution to soil moisture change. Why would there be increased soil moisture when P and ET are changing in the same direction? I have trouble understanding it. Furthermore, if there is increased soil moisture why is Q not increasing? I would not be so quick to jump to this conclusion: perhaps they can take it more slowly, and in steps.

Precipitation and atmospheric demand are both increasing, but at the same time for these same reasons (and other reasons, e.g., temperature) I can understand vegetation activity being increased, which probably removes the increased soil moisture in the root zone due to the increased precipitation (leading to increased ET) but without increased recharge, which keeps Q the same. Not sure if this logic is right.

[Figure]

In any case I am afraid the observed phenomenon may not be fully explained without invoking changes to seasonal variability (of everything, especially NDVI). There must be some kind of nonlinearity caused by changes to the seasonality, which may contribute to the phenomenon. In other words, changes in precipitation and radiation (and wind) propagate through the system in more complex ways than the authors have concluded in the paper. For the present, the paper requires some moderate revisions to address these issues.

I suggest that the authors try to refine their attribution exercise to account for this complex system perspective, and to allow for seasonality changes to play a role in contributing to the phenomenon. A further suggestion, anticipating future studies, is to present a conceptual model (in the form of a causal loop) that possibly accounts for some kinds of feedbacks that may need to be invoked to fully explain the phenomenon. The current paper looks like a stepping stone towards a more comprehensive model of the system in the future.

---

## Referee Comment (RC3) · Anonymous Referee #3 · 7 May 2018

General summary:

In this study, the authors leverage precipitation and streamflow observations to estimate trends in evapotranspiration at over 150 catchments in Austria from 1977 to 2014, and attempt to attribute the trends to numerous potential drivers including: radiation and evaporative demand, vegetation, and water availability. Due to the scarcity of evapotranspiration measurements, detecting and attributing trends in historical evapotranspiration has been difficult. Thus, I think this study is a significant contribution to the current literature, and appreciate the focus on observations. The results suggest that evapotranspiration is generally increasing, and these increases are due to increased

atmospheric demand and radiation, increased vegetation activity, and increased water availability.

Comments:

[1] The authors clearly acknowledge that drivers of ETwb are tightly interlinked in the discussion. Does the attribution methodology adequately take this into account though? Have you explored the covariance among attributing variables? For example, if increases in P lead to increases in NDVI, are increases in P being overstated in the current attribution?

[2] Section 2.1.3. I am a little confused about the time scale of ETwb data being smoothed and plotted in Fig. 2. Are you estimating <annual ETwb> = <annual P> - <annual Q> and smoothing these annual estimates with the Gaussian filter? If so, could increases in precipitation add to storage and not necessarily ET? Also, are the trends in ETwb consistent with changes in ETwb inferred by actually dividing the time series into larger time intervals where changes in storage can assumed to be much smaller (e.g. <ETwb> estimated from average data 1977 to 1995, and <ETwb> estimated from average data 1996 to 2014)?

[3] Lastly, I agree with the three previous comments that the attribution would benefit from a more thorough discussion of the impacts of wind speed and water availability. I think these suggestions were well elaborated in the previous comments. With regards to wind speed though, it could be useful to perform a sensitivity analysis of ETo to possible wind speed trends (e.g. based on the magnitude of the trend inferred from ERA Interim data) rather than using uniform monthly wind speed.

Minor comments:

[1] Page 2 (L26): In addition to $CO_2$, stomata also respond to changes in atmospheric demand, temperature, and soil moisture.

[2] Page 3 (L27): PET, "Potential evapotranspiration".

[Figure]

[3] Page 5 (L14): It may be helpful to explain this in more detail: "wind data were regarded as not representative with respect to evaporation trends", i.e. "not representative" is vague.

[4] Since the phrase "vegetation activity" is used frequently, it might be useful to add a sentence in the 2.3.2 explaining the potential physical mechanisms driving "vegetation activity" (as represented to NDVI), e.g. vegetation fraction, vegetation type, LAI, phenology, etc.

[5] Page 12 (L20): with higher values during the early 1990s, right?

[5] Clarify titles in Figures S7-S8.

---

## Author Comment (AC1) · 1 Jun 2018

**Replies to the comments by Axel Thomas**

We would like to thank Axel Thomas for his interest and for his comments on our manuscript. His main comment addresses the potential effect of changes in wind speeds on evapotranspiration. Two further comments address a statement in the introduction that relates increases in evapotranspiration to global warming, and the use of the term evapotranspiration.

**Potential effect of changes in wind speed**

*There is a vast body of evidence (see in particular the review paper of McVicar et al. (2012, cited) that both radiation and aerodynamic terms are determining potential evapotranspiration (Ep). Depending on region wind may account for the major part of Ep variance. Even though the authors argue with a lack of spatial homogeneity of wind speeds in their study area their averaged wind speeds (Supplement Figure S2b) show quite well the general decrease of wind speeds that has been observed world-wide. The large variability is to be expected in largely mountainous Austria and disregarding this variability does introduce a major error into the analysis. In this respect it is most unfortunate to see that both calculation AND attribution analysis of Penman-Monteith Ep are based on spatially and temporally averaged wind data. Even using averaged wind data as additional variable in the attribution analysis would already show that both radiative and aerodynamic forcing largely explain most of the variance. In the present form – without a realistic inclusion of wind data – the results are misleading. I would propose to recalculate Ep with wind speed data that contains as much spatial and temporal variance as possible. In addition attribution itself is variable both in spatial and temporal terms (Fan and Thomas, https://doi.org/10.1016/j.jhydrol.2018.02.080) so an extended analysis taking into account attribution variability would offer the reader a considerably improved analysis of Ep dynamics.*

Response: We have now analyzed the effect of changes in wind speed. In order to estimate the potential effect of changes in wind speed we derived spatially smoothed patterns of average monthly trends in wind speed from station observations. These were applied to spatial patterns of wind speeds derived from high-resolution downscaled reanalysis data. Initial results show that wind speeds have indeed decreased in Austria (by about 2% per decade) but the effect on trends in reference evapotranspiration is small. When allowing for decreasing wind speed, the average trend in reference evapotranspiration is 2.9% per decade, as compared to 3.1% when assuming no trends in wind speed. The low impact of the changes in wind speed on reference evapotranspiration can be explained by the generally humid climate in Austria, where wind speed has a much lower impact on reference evapotranspiration than in an arid climate (Irmak et al., 2006). We have added the analyses to the supplement and we refer to it in the main text.

**Statement in the introduction**

*There are two smaller points I would also like to mention: the authors rightly point out in their paper that temperature is not an important driver of Ep but at the same time begin their introduction with the much-too–often-heard statement that global warming (hence temperature) has increased regional evapotranspiration. Even the IPCC still voices this scientifically incorrect statement. I would propose to rephrase this sentence to clarify that in the context of global CLIMATIC change Ep also has seen changes.*

Response: We agree and this has been changed as suggested: "In the context of global climatic changes, regional $E$ has increased in many parts of the world in the last decades (Huntington, 2006)."

**Use of the term evapotranspiration**

*Another point to clarify is the sometimes misleading way 'evapotranspiration' is used in this paper. Evapotranspiration is an umbrella term that has many definitions and can be estimated in different ways. In the Introduction most of the papers cited deal with actual evapotranspiration as do the authors when they use the abbreviation 'E' in their data analysis and results. On p 2/l 15 however they appear to mean potential evapotranspiration (at least most of the cited papers deal with Penman-Monteith potential evapotranspiration). On p 3/l 27 it is PET (again perhaps potential evapotranspiration) while reference evapotranspiration (E0, p 6/l 22) is actually crop reference evapotranspiration as the method of Allen et al. 1998 is cited. 'potential evapotranspiration' is used twice in section headlines 2.3 and 2.3.2 but is not defined elsewhere; on p 7/l 23ff 'potential evapotranspiration' and 'reference evapotranspiration' are used almost synonymously. Perhaps the authors might consider adding a short section pointing out the differences between different measures and methods of evapotranspiration cited or used in their paper and then use the appropriate terms consistently throughout their paper.*

Response: Thank you very much for pointing this out. Indeed, the different terms for evapotranspiration have not always been used carefully in the manuscript and we have changed this in the revised version. In detail, we have included the following changes:

P2/L15: Here we discuss potential drivers of changes in actual evapotranspiration, changes in available energy and atmospheric evaporative demand being one of them. Since this may be estimated by pan evaporation, the cited papers in this section deal with pan evaporation.

P3/L27: Reference evapotranspiration has been used for this analysis and this has been changed accordingly.

P6/L22: To our knowledge, both terms, reference evapotranspiration and reference crop evapotranspiration, may be used for the method of Allen et al. (1998).

Headlines 2.3 and 2.3.2: potential evaporation has been replaced by reference evapotranspiration.

P7/L23 ff: All uses of potential evaporation have been changed to reference evapotranspiration.

Reference:

Irmak, S., J.O. Payero, D.L. Martin, A. Irmak, and T.A. Howell (2006): Sensitivity analyses and sensitivity coefficients of standardized daily ASCE-Penman-Monteith equation, *Journal of Irrigation and Drainage Engineering*, *132*(6), 564-578.

---

## Author Comment (AC4) · 1 Jun 2018

**Replies to the comments by Referee #3**

We would like to thank Referee #3 for his/her interest and the comments on our manuscript. These relate to the relationship between the driving variables and the effect on the attribution estimates, the possible effect of variations in storage, the calculation of the trends, and the effect of variations in wind speed.

**Relationship between the driving variables and the effect on the attribution estimates**

*[1] The authors clearly acknowledge that drivers of ETwb are tightly interlinked in the discussion. Does the attribution methodology adequately take this into account though? Have you explored the covariance among attributing variables? For example, if increases in P lead to increases in NDVI, are increases in P being overstated in the current attribution?*

Response: We agree that the effect of increases in $P$ would be overestimated if trends in $P$ and trends in NDVI were correlated and thus Fig. 8c included indirect effects of $P$ on vegetation activity. However, we checked partial correlation coefficients when trends in NDVI and $E_0$ were accounted for, and this did practically not affect the correlation between $P$ and $E_{wb}$. Increases in $P$ are not related to increases in NDVI ($r$ = -0.12) or increases in $E_0$ ($r$ = -0.01) in our study area (Fig. 1). Covariances between $P$ and $E_0$ or NDVI therefore do not influence our attribution analysis. However, Monte Carlo simulations with synthetic $P$ and $Q$ series suggest that the effect of increases in $P$ was overestimated in the current attribution (see the reply to the comments of Ryan Teuling). We take account of this in the revised manuscript.

[Figure]

Fig. 1: Scatter plots of the relationships between trends in $E_0$ and NDVI against the trend in $P$.

**Possible effect of variations in storage**

*[2] Section 2.1.3. I am a little confused about the time scale of ETwb data being smoothed and plotted in Fig. 2. Are you estimating <annual ETwb> = <annual P> - <annual Q> and smoothing these annual estimates with the Gaussian filter? If so, could increases in precipitation add to storage and not necessarily ET?*

Response: Yes, Fig. 2a and d show variations in anomalies of $E_{wb}$ estimated as <annual $E_{wb}$ > = <annual $P$> - <annual $Q$>. Values for individual years cannot be interpreted as variation in $E_{wb}$ since variations in $P$ may have added to storage. The data were therefore smoothed by a Gaussian filter. Changes in surface water, soil, and snow storage can be assumed small over periods of several years. Studies on groundwater level changes in Austria do not show large-scale groundwater changes over the study

period (Blaschke et al., 2011; Neunteufel et al., 2017). We therefore assume that changes in groundwater storage (and changes in any groundwater fluxes) are small. Since changes in glaciers can result in significant storage changes, catchments including glaciers were excluded. This suggests that changes in storage are likely small and that the trend in *P-Q* can be interpreted as trend in *E*.

**Calculation of trends**

*Also, are the trends in ETwb consistent with changes in ETwb inferred by actually dividing the time series into larger time intervals where changes in storage can assumed to be much smaller (e.g. <ETwb> estimated from average data 1977 to 1995, and <ETwb> estimated from average data 1996 to 2014)?*

Response: Trends calculated from the annual series are consistent with changes in $E_{wb}$ derived from dividing the series into two parts from 1977 to 1995 and from 1996 to 2014, as in this equation:

$$t = \frac{\overline{E_{wb96-14}} - \overline{E_{wb77-95}}}{\overline{y_{96-14}} - \overline{y_{77-95}} + 1} * 10$$

Where *t* is the trend (mm $y^{-1}$ decade$^{-1}$), $\overline{E_{wb96-14}}$ ($\overline{E_{wb77-95}}$) is the average $E_{wb}$ over 1996-2014 (over 1977-1995), and $\overline{y_{96-14}}$ ($\overline{y_{77-95}}$) is the average year of the second (first) half of the study period (i.e. 2005 and 1986). Calculating the trend in $E_{wb}$ this way results on average over all catchments in a trend of 30.4 mm $y^{-1}$ decade$^{-1}$, compared to an average trend of 29.3 mm $y^{-1}$ decade$^{-1}$ when calculated from the annual series.

**Effect of variations in wind speed**

*[3] Lastly, I agree with the three previous comments that the attribution would benefit from a more thorough discussion of the impacts of wind speed and water availability. I think these suggestions were well elaborated in the previous comments. With regards to wind speed though, it could be useful to perform a sensitivity analysis of ETo to possible wind speed trends (e.g. based on the magnitude of the trend inferred from ERA Interim data) rather than using uniform monthly wind speed.*

Response: We have now analyzed the effect of changes in wind speed. In order to estimate the potential effect of changes in wind speed we derived spatially smoothed patterns of average monthly trends in wind speed from station observations. These were applied to spatial patterns of absolute wind speeds derived from high-resolution downscaled reanalysis data. This approach was chosen since suggested drivers for the trends in wind speed are changes in the atmospheric circulation and an increase in surface roughness, which however are not captured by reanalysis data (Vautard et al., 2010). Initial results show that wind speeds have indeed decreased in Austria (by about 2% per decade) but the effect on trends in reference evapotranspiration is small. When allowing for decreasing wind speed the average trend in reference evapotranspiration is 2.9% per decade, as compared to 3.1% when assuming no trends in wind speed. We have added the analyses to the supplement and we refer to it in the main text.

**Minor comments:**

*[1] Page 2 (L26): In addition to CO2, stomata also respond to changes in atmospheric demand, temperature, and soil moisture.*

Response: We have reformulated this sentence. Since the focus in this sentence is on the changes in atmospheric $CO_2$ as a further driver of changes in $E$, other factors that influence stomata closure are not mentioned here.

*[2] Page 3 (L27): PET, "Potential evapotranspiration".*

Response: Has been changed to reference evapotranspiration.

*[3] Page 5 (L14): It may be helpful to explain this in more detail: "wind data were regarded as not representative with respect to evaporation trends", i.e. "not representative" is vague.*

Response: Has been changed: "Trends in wind data were not included in the analysis since station observations of wind speeds are known to be prone to inhomogeneities (Böhm, 2008), annual anomalies of wind speed data from 85 stations in Austria appear unrelated to each other (Supplementary Figure S1a), and temporal trends over 1977–2014 do not show any spatial pattern (Supplementary Figure S2a) (see Supplement S1)."

*[4] Since the phrase "vegetation activity" is used frequently, it might be useful to add a sentence in the 2.3.2 explaining the potential physical mechanisms driving "vegetation activity" (as represented to NDVI), e.g. vegetation fraction, vegetation type, LAI, phenology, etc.*

Response: Good idea, we have added the following sentence: "Changes in vegetation activity as observed by the NDVI represent an integrated signal of changes in the phenology, the leaf area index, the vegetation fraction, the vegetation type and the land cover."

*[5] Page 12 (L20): with higher values during the early 1990s, right?*

Response: Yes, thank you.

*[5] Clarify titles in Figures S7-S8.*

Response: Has been changed to "…for biweekly averages over the course of the year (the plot titles indicate the starting day of the two-week period)."

References:

Blaschke, A., Merz, R., Parajka, J., Salinas, J., and Blöschl, G.: Auswirkungen des Klimawandels auf das Wasserdargebot von Grund- und Oberflächenwasser, Österreichische Wasser-und Abfallwirtschaft, 63, 31-41, 2011.

Böhm, R.: Heisse Luft: Reizwort Klimawandel: Fakten, Ängste, Geschäfte, Ed. Va Bene, 2008.

Neunteufel, R., Schmidt, B.-J., and Perfler, R.: Ressourcenverfügbarkeit und Bedarfsplanung auf Basis geänderter Rahmenbedingungen, Österreichische Wasser- und Abfallwirtschaft, 69, 214-224, 2017.

Vautard, R., Cattiaux, J., Yiou, P., Thepaut, J. N., and Ciais, P.: Northern Hemisphere atmospheric stilling partly attributed to an increase in surface roughness, Nature Geoscience, 3, 756-761, 10.1038/ngeo979, 2010.

---

## Author Response (AR1)

**Authors' response - Manuscript "Why has catchment evaporation increased in the past 40 years? A data-based study in Austria" by D. Duethmann and G. Blöschl**

**Replies to the comments by Axel Thomas**

We would like to thank Axel Thomas for his interest and for his comments on our manuscript. His main comment addresses the potential effect of changes in wind speeds on evaporation. Two further comments address a statement in the introduction that relates increases in evaporation to global warming, and the use of the term evaporation.

**Potential effect of changes in wind speed**

*There is a vast body of evidence (see in particular the review paper of McVicar et al. (2012, cited) that both radiation and aerodynamic terms are determining potential evapotranspiration (Ep). Depending on region wind may account for the major part of Ep variance. Even though the authors argue with a lack of spatial homogeneity of wind speeds in their study area their averaged wind speeds (Supplement Figure S2b) show quite well the general decrease of wind speeds that has been observed world-wide. The large variability is to be expected in largely mountainous Austria and disregarding this variability does introduce a major error into the analysis. In this respect it is most unfortunate to see that both calculation AND attribution analysis of Penman-Monteith Ep are based on spatially and temporally averaged wind data. Even using averaged wind data as additional variable in the attribution analysis would already show that both radiative and aerodynamic forcing largely explain most of the variance. In the present form − without a realistic inclusion of wind data − the results are misleading. I would propose to recalculate Ep with wind speed data that contains as much spatial and temporal variance as possible. In addition attribution itself is variable both in spatial and temporal terms (Fan and Thomas, https://doi.org/10.1016/j.jhydrol.2018.02.080) so an extended analysis taking into account attribution variability would offer the reader a considerably improved analysis of Ep dynamics.*

Response: We have now analyzed the effect of changes in wind speed. In a first analysis, we applied average monthly trends derived from station observations of wind speed to the wind speeds used in the original analysis. In a second analysis, we aimed at also including spatial heterogeneities in wind speed and its trends. For this purpose, we derived spatially smoothed patterns of average monthly trends in wind speed from station observations. These were applied to spatial patterns of wind speeds derived from high-resolution downscaled reanalysis data. The results suggest that wind speeds have indeed decreased in Austria (by about 3% per decade) but the effect on trends in reference evaporation is small. According to the first analysis, the trend in $E_0$ averaged over all catchments is 2.4 ± 0.7 % per decade when allowing for decreasing wind speeds, as compared to 2.8 ± 0.7 % per decade when assuming no trends in wind speed. In the second analysis, $E_0$ estimates and trends in wind speed were lower due to lower wind speeds in the reanalysis data compared to the averages of the station data. This led to a smaller effect of the trends in wind speed on $E_0$ than in the first analysis (average $E_0$ trend of 2.9 ± 0.6 % per decade when allowing for trends in wind speed as compared to 3.1 ± 0.6 % per decade when assuming no trends in wind speed). The low impact of the changes in wind speed on reference evaporation can be explained by the generally humid climate in Austria, where wind speed has a much lower impact on reference evaporation than in an arid climate (Irmak et al., 2006). We

have added the analyses to the supplement and we refer to it in sections 2.1.3, 3.2.1 and 4.2 of the main text.

The temporal variability of the drivers of reference evaporation ($E_0$) is already shown in the manuscript (Fig. 3). The spatial variability of the trends in the drivers of $E_0$ is not large and not shown in order to keep the number of figures low.

**Statement in the introduction**

*There are two smaller points I would also like to mention: the authors rightly point out in their paper that temperature is not an important driver of Ep but at the same time begin their introduction with the much-too–often-heard statement that global warming (hence temperature) has increased regional evapotranspiration. Even the IPCC still voices this scientifically incorrect statement. I would propose to rephrase this sentence to clarify that in the context of global CLIMATIC change Ep also has seen changes.*

Response: We agree and this has been changed as suggested: "In the context of global climate change, regional *E* has increased in many parts of the world in the last decades (Huntington, 2006)."

**Use of the term evaporation**

*Another point to clarify is the sometimes misleading way 'evapotranspiration' is used in this paper. Evapotranspiration is an umbrella term that has many definitions and can be estimated in different ways. In the Introduction most of the papers cited deal with actual evapotranspiration as do the authors when they use the abbreviation 'E' in their data analysis and results. On p 2/l 15 however they appear to mean potential evapotranspiration (at least most of the cited papers deal with Penman-Monteith potential evapotranspiration). On p 3/l 27 it is PET (again perhaps potential evapotranspiration) while reference evapotranspiration (E0, p 6/l 22) is actually crop reference evapotranspiration as the method of Allen et al. 1998 is cited. 'potential evapotranspiration' is used twice in section headlines 2.3 and 2.3.2 but is not defined elsewhere; on p 7/l 23ff 'potential evapotranspiration' and 'reference evapotranspiration' are used almost synonymously. Perhaps the authors might consider adding a short section pointing out the differences between different measures and methods of evapotranspiration cited or used in their paper and then use the appropriate terms consistently throughout their paper.*

Response: Thank you very much for pointing this out. Indeed, the different terms for evaporation should be used more carefully and we have changed this in the revised version of the manuscript. We now also consistently use the term evaporation and explain that it includes evaporation from plants via transpiration. Reference evaporation is defined in section 2.3.1. With these two explanations, we found an additional section on the different terms of evaporation not necessary.

In detail, we have included the following changes:

P2/L15: Here we discuss potential drivers of changes in actual evaporation, changes in available energy and atmospheric evaporative demand being one of them. Since this may be estimated by pan evaporation, the cited papers in this section deal with pan evaporation.

P3/L27: Reference evaporation has been used for this analysis and this has been changed accordingly.

P6/L22: To our knowledge, both terms, reference evaporation and reference crop evaporation (as well as reference evapotranspiration and reference crop evapotranspiration), may be used for the method of Allen et al. (1998).

Headlines 2.3 and 2.3.2: potential evapotranspiration has been replaced by reference evaporation.

P7/L23 ff: All uses of potential evaporation have been changed to reference evaporation.

Reference:

Irmak, S., J.O. Payero, D.L. Martin, A. Irmak, and T.A. Howell (2006): Sensitivity analyses and sensitivity coefficients of standardized daily ASCE-Penman-Monteith equation, *Journal of Irrigation and Drainage Engineering*, *132*(6), 564-578.
* * *
**Replies to the comments by Ryan Teuling**

We thank Ryan Teuling for his interest in our study and for his review of our manuscript. The comments of Ryan Teuling concern the potential effect of variations in wind speed, the strong control of $P$ on $E$ trends, and the interrelation between trends in vegetation and trends in soil moisture.

**Potential effect of changes in wind speed**

*Concerning the effect of wind: the potential impact should be discussed in more detail. Wind speed is known to have seen significant trends in many regions (Vautard et al., Nature Geosci. 3, 756-761), and this could have impacted ET trends also in this study. According to the offline PM-equation used by the authors, the impact of wind is direct. It should be noted, however, that when coupled to an atmospheric model, the sensitivity of PM ET for wind becomes much smaller (see e.g. Van Heerwaarden et al., Geophys. Res. Lett. 37, L21401). So I consider it unlikely that wind is a strong driver of ET trends locally, but I agree with the other referee that this warrants an in-depth discussion.*

Response: We have now analyzed the effect of changes in wind speed. In a first analysis, we applied average monthly trends derived from station observations of wind speed to the wind speeds used in the original analysis. In a second analysis, we aimed at also including spatial heterogeneities in wind speed and its trends. For this purpose, we derived spatially smoothed patterns of average monthly trends in wind speed from station observations. These were applied to spatial patterns of wind speeds derived from high-resolution downscaled reanalysis data. The results suggest that wind speeds have indeed decreased in Austria (by about 3% per decade) but the effect on trends in reference evaporation is small. According to the first analysis, the trend in $E_0$ averaged over all catchments is 2.4 ± 0.7 % per decade when allowing for decreasing wind speeds, as compared to 2.8 ± 0.7 % per decade when assuming no trends in wind speed. In the second analysis, $E_0$ estimates and trends in wind speed were lower due to lower wind speeds in the reanalysis data compared to the averages of the station data. This led to a smaller effect of the trends in wind speed on $E_0$ than in the first analysis (average $E_0$ trend of 2.9 ± 0.6 % per decade when allowing for trends in wind speed as compared to 3.1 ± 0.6 % per decade when assuming no trends in wind speed). We have added the analyses to the supplement and we refer to it in sections 2.1.3, 3.2.1 and 4.2 of the main text.

**Strong control of *P* on *E* trends, interrelation between trends in vegetation and trends in soil moisture**

*My main concern related to the interpretation of Figure 8c. This figure shows the relation between inferred trends in P and ET. It suggests a very strong control of P on ET trends, which seems somewhat suspicious given the general humid climate conditions in Austria. In my view, two possible explanations exist.*

Response: We thank the reviewer for raising this point. We agree that the estimated sensitivity of trends in *E* to trends in annual *P* as derived from Fig. 8c is relatively high for a generally humid region.

One reason why we expect a relatively high sensitivity of changes in *E* to changes in *P* in our study is the seasonality of the observed changes in *P*. *P* increased not in a uniform way over the entire year but the increases in *P* were concentrated in the summer season (Supplementary Figure S7 and S8). Thus, the estimated sensitivity is approximately an estimate of the sensitivity of changes in *E* to changes in summer *P*, which can be expected higher than the sensitivity to changes in annual *P*. While changes in summer *P* are expected to contribute more strongly to changes in *E*, changes in winter *P* more likely result in changes in discharge. We discuss this in section 4.2 of the revised manuscript.

However, we carefully rethought the analysis and became aware that the sensitivity derived from the regression in Fig. 8c might be overestimated since water balanced derived *E* (calculated as *P* - *Q*) and *P* are not independent variables. We have performed Monte Carlo simulations with correlated annual *P* and *Q* series to estimate the magnitude of this effect. This analysis aimed at investigating the strength of the relationship between trends in *P* and trends in *Q* resulting from the dependency of the two variables when assuming that trends in *E* are independent of trends in *P*. Means and standard deviations of *P* and *Q*, the covariance between *P* and *Q*, and the spatial variability of the trends in *P* have been derived from the data. This results in regression relationships with a slope of 0.08 ± 0.03 (i.e. 1 mm $y^{-2}$ increase in *P* is related to an increase of *E* by 0.08 ± 0.03 mm $y^{-2}$) and a correlation coefficient of 0.06 ± 0.04. Based on these results, we estimate that the slope derived from Fig. 8c overestimates the sensitivity of changes in *E* to changes in *P* by 0.08 ± 0.03. In the revised paper, we consider this by subtracting the value derived by the Monte Carlo analysis from the regression slope derived from Fig. 8c. This reduces the sensitivity of changes in *E* to changes in *P* from 0.30 ± 0.04 to 0.22 ± 0.05. Consequently, we have revised the attribution. The revised estimates suggest that changes in atmospheric conditions, vegetation activity, and precipitation have contributed 43 ± 15 %, 34 ± 14 %, and 24 ± 5 %, respectively, to the average increase in $E_{wb}$ in the study catchments.

*It might be that in general, soil moisture constraints on ET have weakened because of increased P. In this case, one would expect inferred actual ET to be significantly lower than potential ET. This relation between actual and potential ET, however, is not explored in the manuscript. I believe such an analysis should be included in a revised version, as it provides important insight into the possible background of the drivers of ET trends.*

Response: An analysis of the ratio between $E_{wb}$ to $E_0$ shows ratios close to or even above unity, indicating generally humid conditions. However, estimates of the AET/PET ratio based on the ratio of $E_{wb}$ to $E_0$ likely overestimate the AET/PET ratio. $E_{max}$ (the maximum possible evaporation under the actual vegetation when soils are saturated) is likely much higher than $E_0$. The land cover in the study catchments is dominated by forest and grassland, with average fractions over all study catchments of 0.52 and 0.25. Analyses from non-weighable lysimeters indicate that $E_{max}$ for sites with non-deciduous

trees (pine forests) was about 20-30% higher than $E_0$ (ATV-DVWK, 2001). We estimated $E_{max}$ for each catchment as $E_{max} = E_0 \cdot \sum(l_i \cdot f_i)$, where $l_i$ is the fraction of land cover $i$ and $f_i$ is the ratio of $E_{max}/E_0$ for land cover type $i$, which was approximated as 1.2 for forests and 1 for all other land cover types. This results in median (upper/lower quantile) values for $E_{wb}/E_{max}$ of 0.84 (0.77/0.91), suggesting that in most catchments the AET/PET ratio is <1, even though the study area is classified as humid.

It should be noted that uncertainties in the estimated $E_{wb}$ contribute to uncertainties in the estimated AET/PET ratio. These uncertainties arise to a large part from uncertainties in $P$, e.g. due to undercatch errors and the uncertainties in their correction. While the effect of the undercatch correction on the estimated trends in $E_{wb}$ is small, it has a relatively high influence on uncertainties of the absolute $E_{wb}$ estimates (see section 3.1; table 3).

The estimates of the $E_{wb}/E_{max}$ ratios have been added to the end of section 3.1 of the revised manuscript.

*It should be noted that trends in soil moisture and vegetation might possibly be related. This needs to be discussed, along with the implications for the results shown in Figure 9 which assumes soil moisture/P and vegetation effects to be independent.*

Response: Regarding the interrelation between trends in vegetation and trends in soil moisture, an analysis of the covariance between trends in $P$ and trends in NDVI showed no significant relationship ($r$ = -0.01). This suggests that, in the study area, increases in $P$ were not an important driver for the changes in vegetation activity, and that increases in NDVI are rather driven by increases in air temperature and a longer growing season, increases in atmospheric $CO_2$ and land cover changes (such as the increase in forest at the expense of grassland). We have added this point to section 3.2.4 and to the discussion section of the revised manuscript.

*A second explanation for the relation in Fig. 8c could be that trends in ET are induced by overestimation of trends in P, for instance due to a too strong correction for undercatch. This possibility needs to be explored and discussed.*

Response: An overestimation of trends in $P$ by a too strong correction for undercatch has only a small influence on the estimated relationship between trends in $P$ and trends in $E_{wb}$. As shown in Table 3, the effect of the applied undercatch correction on average trends in $P$ and $E_{wb}$ is small (in contrast to the effect on absolute values). This is also reflected by a small effect on the estimated regression slope between trends in $P$ and trends in $E_{wb}$ and the attribution result. When considering no undercatch correction of precipitation 5.9 ± 1.8 mm y$^{-1}$ decade$^{-1}$ of the $E_{wb}$ trend is estimated to be due to increases in precipitation, as compared to 6.9 ± 1.6 mm y$^{-1}$ decade$^{-1}$ when correcting precipitation for undercatch using parameters for moderately sheltered locations.

*So in summary, if the correlation is physical/causal, the authors should provide additional evidence for the underlying process, for instance by showing increasing ET/PET ratios. In addition, the dependency of trends in vegetation and soil moisture needs to be explored. Fig 9 is interesting, but these results are currently not sufficiently robust to be published.*

Response: We revised the analysis taking into account the dependency between trends in $P$ and trends in $E_{wb}$, analyzed the relationship between trends in vegetation and trends in $P$, and provided additional explanation for a high sensitivity of changes in $E$ to changes in $P$ in the study region. In the revised paper, we formulate the attribution more cautious and more clearly mention the uncertainties.

Despite the remaining uncertainties, we believe that presenting the results in Fig. 9 is a useful contribution.

Reference:

ATV-DVWK (2001): Verdunstung in Bezug zu Landnutzung, Bewuchs und Boden, GFA-Ges. zur Förderung d. Abwassertechnik e.V.
* * *
**Replies to the comments by Referee #2**

We would like to thank Referee #2 for his/her interest and the useful comments on our manuscript. These were related to the attribution to soil moisture change, changes in seasonality and feedback effects between the drivers.

**Attribution to soil moisture change**

*The authors explain the dependence of the ET change (partly) on change of precipitation is as a result of increased soil moisture arising from the precipitation change. Here is where I have some concern - I have trouble grasping the attribution to soil moisture change. Why would there be increased soil moisture when P and ET are changing in the same direction? I have trouble understanding it. Furthermore, if there is increased soil moisture why is Q not increasing? I would not be so quick to jump to this conclusion: perhaps they can take it more slowly, and in steps. Precipitation and atmospheric demand are both increasing, but at the same time for these same reasons (and other reasons, e.g., temperature) I can understand vegetation activity being increased, which probably removes the increased soil moisture in the root zone due to the increased precipitation (leading to increased ET) but without increased recharge, which keeps Q the same. Not sure if this logic is right.*

Response: Thank you for this comment. It made us aware that using soil moisture is not suited well as one of the drivers for the attribution because of the feedbacks between changes in *E* and changes in soil moisture and because it neglects changes in interception. The feedbacks between changes in *E* and changes in soil moisture lead to some ambiguity. For example, under conditions of increasing *E* due to increasing atmospheric demand, *E* increases more strongly at a location with increasing precipitation compared to a location with stationary precipitation but it is unclear whether this effect should be ascribed to changes in soil moisture (as done in the original manuscript). In the revised manuscript, we use precipitation instead of soil moisture as one of the drivers for the attribution analysis.

**Changes in seasonality**

*In any case I am afraid the observed phenomenon may not be fully explained without invoking changes to seasonal variability (of everything, especially NDVI). There must be some kind of nonlinearity caused by changes to the seasonality, which may contribute to the phenomenon. In other words, changes in precipitation and radiation (and wind) propagate through the system in more complex ways than the authors have concluded in the paper. For the present, the paper requires some moderate revisions to address these issues. I suggest that the authors try to refine their attribution exercise to account for this complex system perspective, and to allow for seasonality changes to play a role in contributing to the phenomenon.*

Response: Seasonality effects are partly already considered in the analysis. $E_0$ was calculated on a daily basis considering daily inputs of global radiation, air temperature, and vapor pressure deficit. Thus, variable rates of increase in these inputs during different seasons and their variable effects on $E_0$ during different seasons are considered in the analysis. We investigated changes in NDVI over the year and these were considered in the analysis of changes in $E_{0v}$. With respect to the analysis of the water balance components, the manuscript already showed seasonal changes of precipitation (for the summer and winter half year; Supplementary Figure S7 and S8).

We have now also analyzed changes in *P-Q* and discharge on a seasonal basis (Fig. R 1). *P-Q* shows increases during the summer half year (May-Oct) and decreases over the winter half year (Nov-Apr). Precipitation increases during the summer half year but shows no trends or decreases over the winter half year. Discharge does not show trends over the summer or the winter half year.

Due to intraannual storage variations *P-Q* for the winter or summer half year cannot be interpreted as *E* in the winter or summer half year. Changes in *P-Q* represent a combination of changes in *E* and changes in storage. The negative trend in *P-Q* during the winter half year suggests an increase in evaporation during the winter half year and/or a lower transfer of stored water from the winter to the summer half year. One possible explanation for a lower transfer of stored water might be the decrease in snow, i.e. a greater proportion of the precipitation that falls during winter contributes to discharge during the winter instead of being stored as snow and contributing to discharge or *E* during the summer half year.

**Consideration of feedbacks**

*A further suggestion, anticipating future studies, is to present a conceptual model (in the form of a causal loop) that possibly accounts for some kinds of feedbacks that may need to be invoked to fully explain the phenomenon. The current paper looks like a stepping stone towards a more comprehensive model of the system in the future.*

Response: We agree with the referee about the importance of considering changes in $E_{wb}$ within a systems approach since the changes in *E*, vegetation, soil moisture, etc. are related through multiple feedbacks (Fig. R 2). We now include a causal loop diagram that visualizes these feedbacks and supports the discussion section of our paper. We explicitly discuss which feedbacks are included or excluded with the different drivers in the attribution.

[Figure]

Fig. R 1: Anomalies of (a, d, h, k) *P-Q*, (b, e, i, l) precipitation and (c, f, j, m) discharge for (a-g) summer half years and (h-m) winter half years over 1977–2014. (a–c and h–j) show anomalies by region. Data smoothed using a Gaussian filter with a standard deviation of 2 years. (d–f and k–m) show anomalies over all catchments. The thin blue line shows the mean over all catchments, the grey shaded area the variability between catchments (± 1 standard deviation), the bold black line the smoothed mean, and the red dashed line the trend.

[Figure]

Fig. R 2: Drivers of changes in evaporation, *E*, including feedback effects.
* * *
**Replies to the comments by Referee #3**

We would like to thank Referee #3 for his/her interest and the comments on our manuscript. These relate to the relationship between the driving variables and the effect on the attribution estimates, the possible effect of variations in storage, the calculation of the trends, and the effect of variations in wind speed.

**Relationship between the driving variables and the effect on the attribution estimates**

*[1] The authors clearly acknowledge that drivers of ETwb are tightly interlinked in the discussion. Does the attribution methodology adequately take this into account though? Have you explored the covariance among attributing variables? For example, if increases in P lead to increases in NDVI, are increases in P being overstated in the current attribution?*

Response: We agree that the effect of increases in *P* would be overestimated if trends in *P* and trends in NDVI were correlated and thus Fig. 8c included indirect effects of *P* on vegetation activity. We checked partial correlation coefficients when trends in NDVI and $E_0$ were accounted for, and this did practically not affect the correlation between *P* and $E_{wb}$. In the study region, increases in *P* are not related to increases in NDVI ($r$ = -0.01) or increases in $E_0$ ($r$ = -0.12) (Fig. R 3). The attribution analysis is therefore not influenced by covariances between trends in *P* and $E_0$ or *P* and NDVI. We added the analysis of correlations between trends in *P* and $E_0$ or *P* and NDVI to section 3.2.4 of the manuscript.

However, the effect of increases in *P* on increases in $E_{wb}$ was overestimated due to dependencies between the *P* and $E_{wb}$ series. This is accounted for in the revised manuscript (see the reply to the comments of Ryan Teuling).

[Figure]

Fig. R 3: Scatter plots of the relationships between trends in $E_0$ and NDVI against the trend in $P$.

**Possible effect of variations in storage**

*[2] Section 2.1.3. I am a little confused about the time scale of ETwb data being smoothed and plotted in Fig. 2. Are you estimating <annual ETwb> = <annual P> - <annual Q> and smoothing these annual estimates with the Gaussian filter? If so, could increases in precipitation add to storage and not necessarily ET?*

Response: Yes, Fig. 2a and d show variations in anomalies of $E_{wb}$ estimated as <annual $E_{wb}$ > = <annual $P$> - <annual $Q$>. Values for individual years cannot be interpreted as variation in $E_{wb}$ since variations in $P$ may have added to storage. The data were therefore smoothed by a Gaussian filter. Changes in surface water, soil, and snow storage can be assumed small over periods of several years. Studies on groundwater level changes in Austria do not show large-scale groundwater changes over the study period (Blaschke et al., 2011; Neunteufel et al., 2017). We therefore assume that changes in groundwater storage (and changes in any groundwater fluxes) are small. Since changes in glaciers can result in significant storage changes, catchments including glaciers were excluded. This suggests that changes in storage are likely small and that the trend in $P$-$Q$ can be interpreted as trend in $E$.

**Calculation of trends**

*Also, are the trends in ETwb consistent with changes in ETwb inferred by actually dividing the time series into larger time intervals where changes in storage can assumed to be much smaller (e.g. <ETwb> estimated from average data 1977 to 1995, and <ETwb> estimated from average data 1996 to 2014)?*

Response: Trends calculated from the annual series are consistent with changes in $E_{wb}$ derived from dividing the series into two parts from 1977 to 1995 and from 1996 to 2014, as in this equation:

$$t = \frac{\overline{E_{wb96-14}} - \overline{E_{wb77-95}}}{\overline{y_{96-14}} - \overline{y_{77-95}} + 1} * 10$$

where $t$ is the trend (mm y$^{-1}$ decade$^{-1}$), $\overline{E_{wb96-14}}$ ($\overline{E_{wb77-95}}$) is the average $E_{wb}$ over 1996-2014 (over 1977-1995), and $\overline{y_{96-14}}$ ($\overline{y_{77-95}}$) is the average year of the second (first) half of the study period (i.e. 2005 and 1986). Calculating the trend in $E_{wb}$ this way results on average over all catchments in a trend of 30.4 mm y$^{-1}$ decade$^{-1}$, compared to an average trend of 29.3 mm y$^{-1}$ decade$^{-1}$ when calculated from the annual series and using Sen's slope as in the manuscript.

**Effect of variations in wind speed**

*[3] Lastly, I agree with the three previous comments that the attribution would benefit from a more thorough discussion of the impacts of wind speed and water availability. I think these suggestions were well elaborated in the previous comments. With regards to wind speed though, it could be useful to perform a sensitivity analysis of ETo to possible wind speed trends (e.g. based on the magnitude of the trend inferred from ERA Interim data) rather than using uniform monthly wind speed.*

Response: We have now analyzed the effect of changes in wind speed. In a first analysis, we applied average monthly trends derived from station observations of wind speed to the wind speeds used in the original analysis. In a second analysis, we aimed at also including spatial heterogeneities in wind speed and its trends. For this purpose, we derived spatially smoothed patterns of average monthly trends in wind speed from station observations. These were applied to spatial patterns of wind speeds derived from high-resolution downscaled reanalysis data. This approach was chosen since suggested drivers for the trends in wind speed are changes in the atmospheric circulation and an increase in surface roughness, which however is not captured by reanalysis data (Vautard et al., 2010). The results suggest that wind speeds have indeed decreased in Austria (by about 3% per decade) but the effect on trends in reference evaporation is small. According to the first analysis, the trend in $E_0$ averaged over all catchments is 2.4 ± 0.7 % per decade when allowing for decreasing wind speeds, as compared to 2.8 ± 0.7 % per decade when assuming no trends in wind speed. In the second analysis, $E_0$ estimates and trends in wind speed were lower due to lower wind speeds in the reanalysis data compared to the averages of the station data. This led to a smaller effect of the trends in wind speed on $E_0$ than in the first analysis (average $E_0$ trend of 2.9 ± 0.6 % per decade when allowing for trends in wind speed as compared to 3.1 ± 0.6 % per decade when assuming no trends in wind speed). We have added the analyses to the supplement and we refer to it in sections 2.1.3, 3.2.1 and 4.2 of the main text.

**Minor comments:**

*[1] Page 2 (L26): In addition to CO2, stomata also respond to changes in atmospheric demand, temperature, and soil moisture.*

Response: We have reformulated this sentence to "Rising atmospheric $CO_2$ concentrations increase plant growth and  influence stomata closure (Gedney et al., 2006)." Since the focus in this sentence is on the changes in atmospheric $CO_2$ as a further driver of changes in *E*, other factors that influence stomata closure are not mentioned here.

*[2] Page 3 (L27): PET, "Potential evapotranspiration".*

Response: Has been changed to reference evaporation.

*[3] Page 5 (L14): It may be helpful to explain this in more detail: "wind data were regarded as not representative with respect to evaporation trends", i.e. "not representative" is vague.*

Response: The sentence has been changed to "Trends in wind data were not included in the analysis since station observations of wind speeds are known to be prone to inhomogeneities (Böhm, 2008), annual anomalies of wind speed data from 85 stations in Austria appear unrelated to each other

(Supplementary Figure S1a), and temporal trends over 1977–2014 do not show any spatial pattern (Supplementary Figure S2a) (see Supplement S1)."

*[4] Since the phrase "vegetation activity" is used frequently, it might be useful to add a sentence in the 2.3.2 explaining the potential physical mechanisms driving "vegetation activity" (as represented to NDVI), e.g. vegetation fraction, vegetation type, LAI, phenology, etc.*

Response: Good idea. We have added the following sentence: "Changes in vegetation activity as observed by the NDVI represent an integrated signal of changes in the phenology, the leaf area index, the vegetation fraction, the vegetation type and the land cover."

*[5] Page 12 (L20): with higher values during the early 1990s, right?*

Response: Yes, thank you.

*[5] Clarify titles in Figures S7-S8.*

Response: Has been changed to "…for biweekly averages over the course of the year (the plot titles indicate the starting day of the two-week period)."

[Figure]

**Supplementary Figure S 3: Anomalies of $E_0$ (a, c) considering trends in wind speed (b, d) with wind speeds as of 1994 for all years. (a, b) refers to an analysis that applied average monthly relative trends to the wind speeds used in the original study. (c, d) refers to a second analysis that considered spatial heterogeneities in wind speed and its trends. The thin blue line shows the mean over all catchments, the grey shaded area shows the variability between catchments (± 1 standard deviation), the bold black line shows the filtered mean (10-year Gauss filter with a standard deviation of 2 years), and the dashed red line the linear trend.**

[Figure]

**Supplementary Figure S 4: Mean contributions of variations in net radiation (R), air temperature (T), vapor pressure deficit (vpd), wind speed (u), their two-way interaction effects and all three way interaction effects (3-way) to the trend in $E_0$. Bars show means over all catchments, error bars show the standard deviation of the variation between catchments. Percent are relative to trends in $E_0$. (a) refers to an analysis that applied average monthly relative trends to the wind speeds used in the original study, (b) refers to a second analysis that considered spatial heterogeneities in wind speed and its trends.**

Supplement S3

**Monte Carlo simulations for estimating the overestimation of the regression relationship between trends in precipitation and trends in evaporation**

Since $E_{wb}$ is estimated from precipitation ($P$) and discharge ($Q$), trends in $E_{wb}$ are not independent from trends in $P$ and a regression relationship between these two variables may overestimate the effect of trends in $P$ on trends in $E_{wb}$. We therefore performed Monte Carlo simulations that aimed at investigating the strength of the relationship between trends in $P$ and trends in $Q$ resulting from the dependency of the two variables when assuming that trends in $E$ are independent of trends in $P$. The results depend on the assumed statistical properties of the data, amongst others on the spatial variability of the trend in $P$ (the stronger the spatial variability of the $P$ trend, the weaker the relationship when assuming trends in $E$ independent of trends in $P$). For the Monte Carlo simulations, we generated $n$ correlated, normal distributed series of annual $P$ and annual $Q$ (Supplementary Figure S 5). Means and standard deviations of annual $P$ and $Q$, and the covariance between annual $P$ and $Q$ were set according to the data set of this study, and $n$ was set to the number of study catchments. Trends in $P$ were considered by adding linear trends to the $P$ series. The variability of the trends between catchments was assumed normal distributed and the mean and variability were derived from the study data. In accordance with the assumption that trends in $E$ are independent of trends in $Q$, the trend added to the $P$ series was also added to the associated $Q$ series. Annual $E$ series were calculated as $P$ minus $Q$. Trends in the $E$ and $P$ series were estimated using Sen's slope. We performed a linear regression between trends in $E$ and trends in $P$ over the $n$ data points and calculated the slope and the coefficient of determination. This procedure was repeated $m=1000$ times and the mean and standard deviation of the slope and the coefficient of determination over these $m$ repetitions were calculated.

The resulting regression relationships have a slope of $0.08 \pm 0.03$ (mean ± standard deviation) and a correlation coefficient of $0.06 \pm 0.04$, suggesting that the slope derived from the regression of $E_{wb}$ against $P$ overestimates the sensitivity of changes in $E$ to changes in $P$ by $0.08 \pm 0.03$. The sensitivity of the trend in $E_{wb}$ to trends in $P$ was therefore estimated as the slope of the linear regression between trends in $E_{wb}$ and trends in $P$ corrected by this value.

**Supplementary Figure S 5: Estimating the overestimation of the regression relationship between trends in precipitation and trends in evaporation, caused by the dependency of the precipitation and evaporation series, by Monte Carlo simulations.**

**Further supplementary tables and figures**

[revised manuscript text omitted]

**Supplementary Figure S 10** Scatterplots of catchment average NDVI (*y*-axis) versus median catchment elevation (*x*-axis) for biweekly averages over the course of the year (the plot titles indicate the starting day of the two-week period).

[Figure]

**Supplementary Figure S 11 Scatterplots of catchment average NDVI trend over 1982–2014 (*y*-axis) versus median catchment elevation (*x*-axis) for biweekly averages over the course of the year (the plot titles indicate the starting day of the two-week period).**